# Functional and evolutionary significance of unknown genes from uncultivated taxa

Álvaro Rodríguez del Río[1], Joaquín Giner-Lamia[1,2,9], Carlos P. Cantalapiedra[1], Jorge Botas[1], Ziqi Deng[1], Ana Hernández-Plaza[1], Martí Munar-Palmer[1], Saray Santamaría-Hernando[1], José J. Rodríguez-Herva[1,2], Hans-Joachim Ruscheweyh[3], Lucas Paoli[3], Thomas S. B. Schmidt[4], Shinichi Sunagawa[3], Peer Bork[4,5,6], Emilia López-Solanilla[1,2], Luis Pedro Coelho[7,8,10] & Jaime Huerta-Cepas[1✉]

Many of the Earth's microbes remain uncultured and understudied, limiting our understanding of the functional and evolutionary aspects of their genetic material, which remain largely overlooked in most metagenomic studies[1]. Here we analysed 149,842 environmental genomes from multiple habitats[2–6] and compiled a curated catalogue of 404,085 functionally and evolutionarily significant novel (FESNov) gene families exclusive to uncultivated prokaryotic taxa. All FESNov families span multiple species, exhibit strong signals of purifying selection and qualify as new orthologous groups, thus nearly tripling the number of bacterial and archaeal gene families described to date. The FESNov catalogue is enriched in clade-specific traits, including 1,034 novel families that can distinguish entire uncultivated phyla, classes and orders, probably representing synapomorphies that facilitated their evolutionary divergence. Using genomic context analysis and structural alignments we predicted functional associations for 32.4% of FESNov families, including 4,349 high-confidence associations with important biological processes. These predictions provide a valuable hypothesis-driven framework that we used for experimental validatation of a new gene family involved in cell motility and a novel set of antimicrobial peptides. We also demonstrate that the relative abundance profiles of novel families can discriminate between environments and clinical conditions, leading to the discovery of potentially new biomarkers associated with colorectal cancer. We expect this work to enhance future metagenomics studies and expand our knowledge of the genetic repertory of uncultivated organisms.

Over the past few decades, metagenomics and metabarcoding studies have revolutionized microbial genomics, not only discovering numerous new bacterial and archaeal lineages[7,8] but also unveiling their unique genetic repertoire[9–11]. The fraction of environmental genes lacking homologues in cultured organisms, hereafter referred to as 'unknown', varies from approximately 25% (ref. 6) to nearly 50% (ref. 3) depending on the methodology and specific environment. This translates to millions of novel genes whose functional aspects and ecological and evolutionary significance remain unknown.

This uncharted microbial sequence space can harbour new genes of great biological significance[10,12,13], including those that may have facilitated the emergence and diversification of major lineages[14] and functional innovations such as novel enzymes[15], antibiotics[16], versatile small peptides[17] and metabolic pathways[4]. However, the systematic curation and analysis of unknown genes at a large scale remains challenging. The genetic pool of uncultivated taxa is currently unrepresented in reference databases of gene functions, protein domains and orthologous groups, limiting our analyses and intercommunity comparisons to known genes derived from cultured organisms.

To date, this shortcoming has primarily been due to technical constraints that have been overcome only recently. First, the task of clustering genes into families and conducting homology searches on a metagenomics scale, a crucial initial step for any comparative analysis, has been rendered achievable only by recent breakthroughs[18,19]. Second, many comparative genomic analyses depend on the availability of complete genomic assemblies and have become feasible only with

---

[1]Centro de Biotecnología y Genómica de Plantas, Universidad Politécnica de Madrid (UPM) – Instituto Nacional de Investigación y Tecnología Agraria y Alimentaria (INIA-CSIC), Madrid, Spain. [2]Departamento de Biotecnología-Biología Vegetal, Escuela Técnica Superior de Ingeniería Agronómica, Alimentaria y de Biosistemas, Universidad Politécnica de Madrid (UPM), Madrid, Spain. [3]Department of Biology, Institute of Microbiology and Swiss Institute of Bioinformatics, ETH Zürich, Zürich, Switzerland. [4]Structural and Computational Biology Unit, European Molecular Biology Laboratory, Heidelberg, Germany. [5]Max Delbrück Centre for Molecular Medicine, Berlin, Germany. [6]Department of Bioinformatics, Biocenter, University of Würzburg, Würzburg, Germany. [7]Institute of Science and Technology for Brain-Inspired Intelligence, Fudan University, Shanghai, China. [8]MOE Key Laboratory of Computational Neuroscience and Brain-Inspired Intelligence, and MOE Frontiers Center for Brain Science, Shanghai, China. [9]Present address: Departamento de Bioquímica Vegetal y Biología Molecular, Facultad de Biología, Instituto de Bioquímica Vegetal y Fotosíntesis (IBVF), Universidad de Sevilla-CSIC, Seville, Spain. [10]Present address: Centre for Microbiome Research, School of Biomedical Sciences, Queensland University of Technology, Translational Research Institute, Woolloongabba, Queensland, Australia. ✉e-mail: j.huerta@csic.es

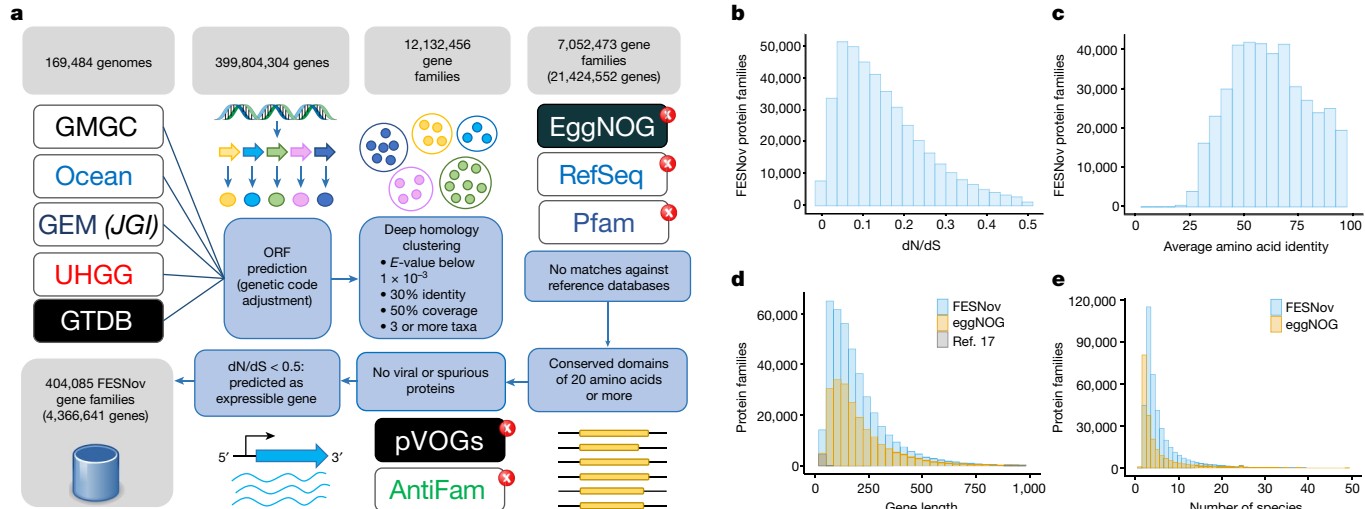

**Fig. 1 | Gene family discovery pipeline and general statistics. a**, Workflow used to identify FESNov gene families. ORF, open reading frame. **b**, dN/dS ratio in FESNov gene families. **c**, Distribution of average protein amino acid identity of FESNov gene families. **d**, Comparison between protein length distribution of FESNov gene families (blue bars), all bacterial and archaeal eggNOG orthologous groups (yellow bars) and the small peptides reported in Sberro et al.[17] (grey bars). **e**, Distribution of species count per family for the FESNov catalogue (blue bars) compared with bacterial and archaeal eggNOG orthologous groups (yellow bars).

the recent publication of hundreds of thousands of environmental genomes[2,3,6].

Several studies have leveraged these advances to successfully harness the concept of gene families for the identification and quantification of genetic novelty on a broad metagenomic scale[3,6,17]. This has enabled the annotation of distant homologues[20], the functional prediction of unknown genes based on coexpression patterns[21] and the identification of lineage-specific novel gene families defining entire new lineages[22]. However, recent work has shown that most genes within the global microbiome can be clustered into a limited number of gene families primarily composed of neutral (or nearly neutral) gene variants[2] (that is, lacking strong positive selection pressures). Additionally, various studies have indicated that both known and unknown gene families are largely dominated by species-specific sequence clusters (orphans) of narrow ecological scope[2,20].

In this study we systematically analysed the genetic repertoire of uncultured bacteria and archaea using a global-scale comparative genomics approach, aiming to identify novel gene families with clear signs of biological significance.

## A curated catalogue of novel gene families

Employing a comparative genomics approach, we analysed a large multihabitat dataset comprising 149,842 medium- and high-quality metagenome-assembled genomes (MAGs) and single-amplified genomes (SAGs), alongside 19,642 reference genomes, from isolated and fully sequenced species. This set includes nearly 400 million gene predictions and was assembled by unification of five data sources representing 82 habitats (Supplementary Table 1): two MAG collections spanning thousands of samples of various origins (GEM[3] and GMGC[2]), a comprehensive human gut catalogue (UHGG[6]), an extensive repository of global oceanic data (OMD)[4] and the GTDB-r95 reference database[5].

To group sequences into gene families we employed a deep-homology-clustering strategy operating at the amino acid level (30% minimum identity, coverage over 50%, expected value (*E*-value) under 0.001). The chosen approach was tailored to maximize the grouping of distantly related genes and has been successfully employed in previous work[2,20,21]. Of the 12,132,456 clusters generated, 58% (7,052,473 clusters) lacked discernible homologues in the databases Pfam-A/B[23], eggNOG v5 (ref. 24) and RefSeq[25]—that is, they represent gene families that are exclusive to uncultivated taxa.

To identify unknown gene family clusters of maximal evolutionary and functional significance (Fig. 1a), we started by selecting clusters that encompassed a minimum of three complete genes from at least two distinct uncultivated species, thereby excluding orphan gene families from our final catalogue. Furthermore, we required gene family members to exhibit a conserved aligned region spanning more than 20 contiguous amino acids. These criteria were important in regard to discarding clusters inferred from fragmented genes and to infer a distinct genomic signature (putative new protein domain) for each novel gene family.

To ensure the exclusion of pseudogene-based and viral-specific gene families, we eliminated clusters that matched either the AntiFam[26] or pVOG[27] databases. In addition, we required the selected novel gene families to exhibit signs of purifying selection (non-synonymous to synonymous substitutions ratio (dN/dS) below 0.5; Fig. 1b), consistent with expectations for functional coding sequences[28]. Moreover, we ascertained their expressibility through either in silico predictions using the software RNAcode[29] or empirical evidence, identifying significant hits against recent metatranscriptome surveys[21,30].

From an evolutionary perspective these novel gene families are conserved, with an average amino acid identity of 62.7% (Fig. 1c). Phylogeny-based orthology prediction[31] suggests that none of the families were involved in gene duplication events at basal taxonomic ranks, probably representing novel orthologous groups at the bacterial or archaeal level. In total, our analysis led to the identification of 404,085 clusters that potentially represent FESNov gene families exclusively from uncultivated taxa. These families exhibit distributions of family size, sequence length and species content comparable to those observed in the eggNOG database (Fig. 1d,e).

In relative terms, the FESNov catalogue constitutes a small (5.7%) subset of the initial total of inferred unknown gene clusters, highlighting the strong impact of quality filters on the analysis of the uncharted sequence space. However, our curated catalogue represents a roughly threefold increase in the total number of prokaryotic orthologous groups known to date (namely, 219,934 bacterial and archaeal eggNOG orthologous groups; Fig. 1d). This underscores the importance of incorporating the genetic repertory of uncultivated taxa into publicly accessible functional genomics repositories.

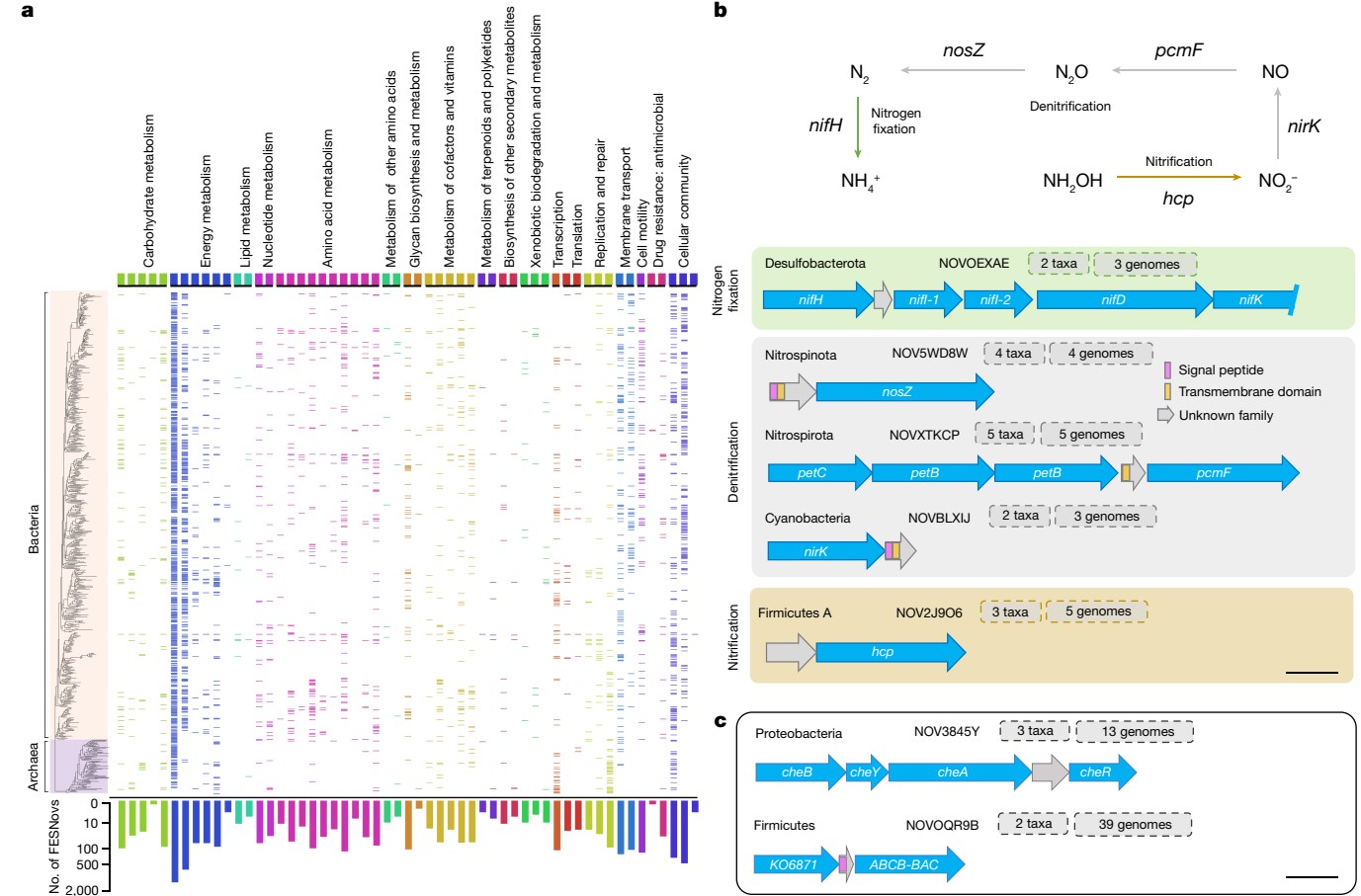

**Fig. 2 | Distribution of FESNov gene families confidently linked to KEGG pathways. a**, Presence/absence matrix of FESNov families associated with KEGG[34] pathways (columns) across the bacterial and archaeal GTDB phylogeny collapsed at the order level (rows) with confidence equal to or greater than 0.9. Taxonomic orders without detections are not shown. The total number of FESNov families per pathway (no. of FESNovs) is shown on a logarithmic scale. FESNov gene families associated with each KEGG pathway can be explored at http://novelfams.cgmlab.org. **b**, Examples of FESNov gene families tightly coupled with genes related to nitrogen cycling. Gene names correspond to: *nifH* (K02588), nitrogenase iron protein; *nifI1* (K02589), nitrogen regulatory protein PII 1; *nifI2* (K02590), nitrogen regulatory protein PII 2; *nifD* (K02586), nitrogenase molybdenum–iron protein alpha chain; *nifk* (K02591), nitrogenase molybdenum–iron protein beta chain; *nosZ* (K00376), nitrous oxide reductase; *petC* (K03886), menaquinol-cytochrome c reductase iron–sulfur subunit; *petB* (K03887), menaquinol-cytochrome c reductase cytochrome b subunit; *pcmF/norC* (K02305), nitric oxide reductase subunit C; *nirK* (K00368), nitrite reductase (NO forming); *hcp* (K05601), hydroxylamine reductase. Respective protein lengths of represented FESNov gene families are 52, 144, 48, 80 and 165. **c**, Genomic context of experimentally validated FESNov gene families NOV3845Y (cell motility) and NOVOQR9B (antimicrobial peptide). Scale bars, 500 base pairs.

## Functional predictions

For exploration of the putative functional roles of FESNov families we used genomic context reconstruction, synteny conservation analysis, structural prediction and detection of intrinsic sequence signals.

Historically, the preservation of gene order across species provided valuable insights into functional interactions—an approach often referred to as the 'guilty-by-association' strategy. Although this technique has proved particularly effective for the accurate prediction of prokaryotic gene functions[32], its efficacy varies across functional categories[33] and may be affected by the quality and completeness of environmental genomes. To account for these issues, we first conducted a benchmarking exercise based on functionally annotated genes to establish a baseline for the minimum level of genomic context conservation required to generate dependable predictions for different Kyoto Encyclopedia of Genes and Genomes (KEGG) pathways[34]. To this end we implemented two scores: (1) 'syntenic conservation', to reflect the conservation of gene order across species and (2) 'functional relatedness of neighbouring genes', measured as the number of contiguous genes belonging to the same KEGG pathway. Results indicate that genomic context analysis has the capacity to

predict accurately (confidence equal to or greater than 0.9) functional associations between genes spanning 55 KEGG pathways. However, the level of support needed in terms of synteny and functional relatedness scores varied among different functional categories (Supplementary Table 2).

Insights from the benchmarking exercise were used to predict KEGG pathway associations for all novel gene families in our catalogue, fine-tuning the thresholds for both synteny and functional relatedness based on the requirements for each pathway. This resulted in a total of 52,793 gene families successfully mapped to at least one specific KEGG pathway with confidence scores exceeding 50%. Of these gene families, 4,349 attained a confidence score of 90% or above. Notably, the highest confidence scores were associated with conserved operon-like regions related to key cellular processes including central metabolism, chemotaxis and degradation pathways (Fig. 2a and Supplementary Table 3). In addition, we identified 17,717 FESNov gene families located in the vicinity of various antibiotic-resistance genes, indicating a possible role in cell defence systems.

Next, we functionally annotated the FESNov catalogue based on each family's predicted structural similarity to functionally known genes. To this end we performed de novo protein structure prediction

for the entire FESNov catalogue using ColabFold[35], which yielded 389,638 protein structures, 226,991 with high-confidence scores (predicted local distance difference test (PLDDT) score equal to or greater than 70). A total of 56,609 FESNov families showed significant structural similarities to known genes in the databases PDB[36] or Uniprot[37]. Interestingly, the same KEGG pathway annotation was predicted for 38.8% of FESNov gene families with both structural homologues and high-confidence (equal to or greater than 90%) genomic context predictions. In addition, 8,804 families had structural similarities to various antibiotic-resistance genes, 3,637 of which had antibiotic-resistance genomic neighbours in their context (Supplementary Table 4).

We also screened FESNov families for inherent sequence features not captured by structural or homology-based searches, including signal peptides, transmembrane regions and potential antimicrobial signals. It was found that 32.9 and 23.7% of FESNov gene families encoded for transmembrane and signal peptide-containing proteins, respectively, suggesting that a significant fraction may play a role in environmental interactions. Additionally, we found 240 short (below 50 amino acid (aa)) FESNov families with antimicrobial signatures (Supplementary Table 5), 17 of which were located in conserved antibiotic-related genomic contexts.

Finally we mapped the FESNov catalogue against several databases that could provide indirect functional hints. Searching through a collection of biosynthetic clusters recently reported from marine metagenomes[4], we found 1,864 matches that may be actively involved in the biosynthesis of natural products (Supplementary Table 6). Mapping FESNov gene families against a set of 11,779 unknown genes recently characterized using genome-wide mutant fitness experiments[38], we found 68 matches to genes associated with specific growth conditions (Supplementary Table 7). Similarly, we found 476 significant hits against a catalogue of potentially functional small peptides[17].

All functional predictions, along with their confidence scores and supporting materials, are available in the online FESNov database (https://novelfams.cgmlab.org). In addition, the FESNov catalogue has been integrated into the eggNOG-mapper tool[39] to facilitate the detection and annotation of novel gene families in any metagenome.

## Hypothesis-driven functional validations

To further validate our functional annotation approach we experimentally tested the functional predictions of two novel gene families, NOV3845Y and NOVOQR9B (Fig. 2c).

NOV3845Y was found in at least two Gram-negative species from the unknown genus *UBA8309* within the class Alphaproteobacteria, and it was predicted to be involved in bacterial chemotaxis (confidence score of 0.8). A low-support protein structure was predicted for this family, and no structural similarity was found to any known gene in PDB or Uniprot. However, given its position in the *che* operon, a gene cluster canonically responsible for chemotaxis, we hypothesized that NOV3845Y played a role in cell motility in response to environmental changes. To test our hypothesis we performed swimming chemotaxis assays on an *Escherichia coli* strain heterologously expressing NOV3845Y and compared it against another *E. coli* strain transformed with the empty vector (pBSK). Results showed that the strain expressing NOV3845Y exhibited a larger swimming halo than the pBSK control strain (Extended Data Fig. 1), confirming the role of NOV3845Y in chemotaxis signalling and cell motility.

NOVOQR9B, a gene family of short peptides found in 39 *Faecalibacillus* genomes, is systematically conserved in a genomic region predicted to be associated with antibiotic resistance (three conserved resistance neighbours within the CARD database). In addition, NOVOQR9B contains antimicrobial signal motifs (antimicrobial probability by Macrel[40], 0.545). We synthesized the 36-amino-acid-long peptide and tested, in vitro, its potential antimicrobial activity against *Paenibacillus polymyxa*, *Bacillus subtilis*, *Lactobacillus* sp., *E. coli* and *Pseudomonas*

*putida* (three Gram-positive and two Gram-negative bacterial species). NOVOQR9B effectively inhibited growth of all three Gram-positive bacterial species but had no inhibitory effect on Gram-negative species (Extended Data Fig. 2).

We also found five FESNov families adjacent to known genes related to nitrogen metabolism, covering operonic regions associated with three major pathways of the nitrogen cycle (Fig. 2b). Among these, NOV5WD8W appeared particularly intriguing because of its direct proximity to a *NosZ*-encoding gene, which is responsible for the reduction of nitrous oxide ($N_2O$) to molecular nitrogen ($N_2$) in the final step of bacterial denitrification. The maturation of *NosZ* requires several accessory proteins, including a copper metal chaperone (*NosL*) involved in copper supply to *NosZ*[41]. Interestingly, NOV5WD8W folds similarly (PLDDT, 82.3; alignment score, 263) to the *E. coli* copper metallochaperone *CusF* (Extended Data Fig. 3), suggesting that this novel gene family could replace *NosL* in the transfer of copper to *NosZ*.

## Density of novel families per genome

Taxonomically, FESNov families are distributed across the entire microbial phylogeny (Fig. 3a) and comprise an important fraction of the genetic repertoire of uncultivated taxa. On average, archaeal genomes encode a larger fraction of FESNov families than bacterial (3.4 versus 1.7%), reflecting the historical delay in characterization of the archaeal domain.

Despite the known widespread phylogenetic distribution of microbial dark matter[20,42] and lineages with a high content of functionally unknown gene families[22], we further interrogated our data to identify potential novelty hubs—that is, lineages with a high number of FESNov families per genome (Supplementary Table 8). At the phylum level, Riflebacteria, known for their high proportion of unidentified genes[43], exhibit the highest co-occurrence of FESNov families per genome (431 ± 171), along with the unknown phylum UBP17 (470 ± 106). Asgardarchaeota, the prokaryotic phylum closest to eukaryotes, also stands out with 228 ± 167 novel gene families per genome (ranked third among phyla). The Patescibacteria phylum, previously reported to contain a high number of unknown genes[20,22,42], exhibits a low novelty density per genome compared with other clades (62 ± 30, 62nd-ranked phylum), although FESNov families represent a significant proportion of their gene set when normalizing by genome size (7.7% ± 2.9 s.d., 13th ranked phylum). Notably, Patescibacteria genomes show a higher proportion (53%) of novel transmembrane and signal peptide proteins than other genomes (33%, two-sided Wilcoxon test $P < 1 \times 10^{-15}$). Transmembrane proteins have been shown to be important for this phylum[22] and, together with secreted proteins involved in cell–cell interactions, they may be relevant because of their probable episymbiotic lifestyle[22,43]. Among these we found several FESNov families in genomic contexts related to both cell wall disruption and DNA degradation and incorporation, indicating that they may indeed play a role in microbe host parasitization[44] (Extended Data Fig. 4).

## Synapomorphies in uncultivated taxa

To identify FESNov families with the greatest evolutionary significance we searched the catalogue for potential synapomorphic traits, an approach previously applied to find evolutionarily relevant unknown genes in specific lineages[14,22,45]. A gene family is considered synapomorphic if it is nearly ubiquitous within a certain clade (high coverage) but mostly absent in other clades (high specificity), therefore indicating common unique derived traits (Methods).

To identify the most prominent synapomorphies in uncultivated lineages we scanned the entire set of 12,132,456 clusters originally inferred with a minimum of 90% clade coverage and 100% clade specificity. Although these thresholds offer very little tolerance for the potential incompleteness of environmental genomes used, and/or horizontal

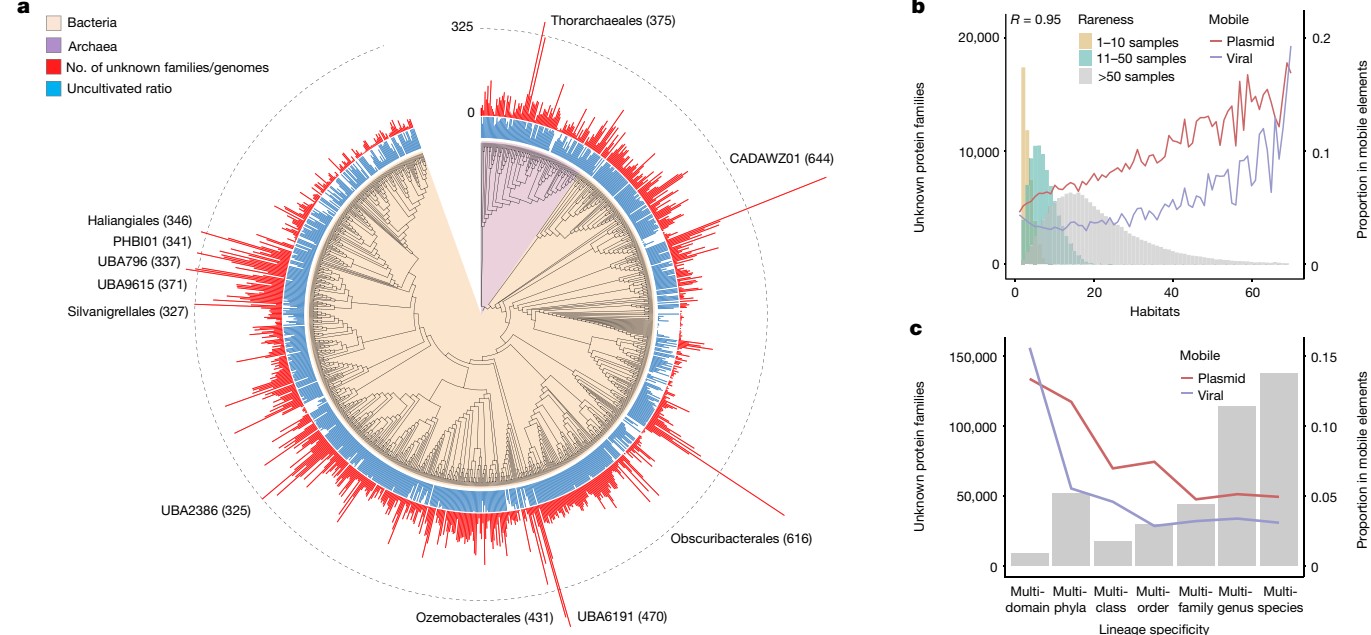

**Fig. 3 | FESNov gene families are spread across the entire microbial phylogeny, covering a variety of habitats. a**, Phylogenetic distribution of FESNov gene families across the GTDB[5] bacterial and archaeal phylogeny, collapsed at the order level. Red bars indicate the number of unknown gene families per genome in each taxonomic order; blue bars represent the proportion of uncultivated species under each collapsed order. Branches with more than 325 unknown gene families are indicated. **b**, Ecological breadth (measured as number of habitats) of FESNov gene families binned into three levels of rareness (number of samples in which they are detected). Red and blue lines indicate the proportion of gene families predicted as mobile in plasmids and viral contigs, respectively. *R* indicates Spearman correlation coefficient between number of habitats and proportion of mobile (in plasmid or viral contigs) FESNov gene families. **c**, Number of FESNov gene families confined to each taxonomic rank attending to the last common ancestor of their family members. Lines indicate the proportion of gene families detected in plasmids (red) or viral (blue) contigs.

gene transfers, we found 2,210 clusters that could be considered synapomorphic at the phylum, class or order level (Supplementary Table 9). Interestingly, almost half of these high-level synapomorphic clusters (1,034 families) were included in our final FESNov catalogue, supporting the idea that our set is greatly enriched in evolutionarily relevant families.

FESNov synapomorphic gene families allow precise distinction of 18 uncultivated phyla, 20 classes and 90 orders, corresponding to 181, 128 and 725 synapomorphic novel gene families, respectively. We propose that these newly identified synapomorphic proteins are probably functional innovations that fostered the ancestral divergence and selection of the underlying lineages. This hypothesis is supported by the fact that these synapomorphic gene families exhibit stronger signs of purifying selection (indicated by a lower dN/dS ratio) and a higher genomic context conservation than other gene families in the FESNov catalogue (Fig. 4b,c). Similarly, the number of unknown synapomorphic gene families was particularly high in uncultivated lineages with poorly understood biology. Examples include the recently proposed Riflebacteria phylum, with 26 putative synapomorphic detections, or the Thorarchaeia and Lokiarchaeia classes, in which 14 detected putative synapomorphic families may provide insight into their divergence from other Asgard members.

For instance, the Thorarchaeia synapomorphic family NOV4IF0P is embedded within a highly conserved genomic region containing eukaryotic-like genes associated with protein translation (*rpl35a* and *elp3*; Fig. 4a). Similarly, we found all genomes within the HRBIN17 class to contain what we believe to be a highly divergent version of *ccmD*, a small protein involved in cytochrome biosynthesis (prediction supported by genomic context analysis). We also found synapomorphic FESNov families in the UBP6 phylum potentially involved in DNA repair (*priA* and *dksA*), or in the Riflebacteria phylum involved in chemotaxis (*mcp* and *dgt*) (Fig. 4a).

## Habitat distribution of novel families

For estimation of the ecological range of FESNov families we mapped their genetic signatures to a larger and standardized set of 63,410 publicly accessible metagenomic samples covering 157 habitats (Supplementary Table 10). Most FESNov gene families (59.4%) were detected in more than ten samples originating from at least two distinct habitats (Supplementary Table 11). The same pattern was found even after consolidation of habitat annotations into ten less detailed ecological groups. For example, we identified various extreme cases in which 243, 16 and two FESNov gene families, respectively, appeared in over half of all marine (3,441 total samples), soil (4,418) and human gut (32,544) samples (Supplementary Table 12).

This result stands in stark contrast to the habitat- and sample-specific patterns seen in most individual species-level genes[2] and functionally unknown gene clusters[20]. This is remarkable considering that none of our original filters used to identify FESNov families included ecological parameters. It also suggests that FESNov families may represent unknown molecular functions from uncultivated microbial lineages spread over geographically and/or ecologically diverse habitats. Alternatively, they may derive from mobile elements.

In an attempt to quantify the contribution of each scenario we identified protein clusters that contained at least one member in a plasmid or viral-like contig. Findings indicated a strong correlation between mobility and the ecological breadth of gene families across different habitats (blue and red lines in Fig. 3b and Extended Data Fig. 5). Similarly, we found that the proportion of mobile elements was correlated with the taxonomic breadth of gene families (Fig. 3c and Extended Data Fig. 6). Nevertheless, the vast majority of families (90.4%) appear detached from obvious events of horizontal gene transfer, suggesting a more constitutive role in their host genomes.

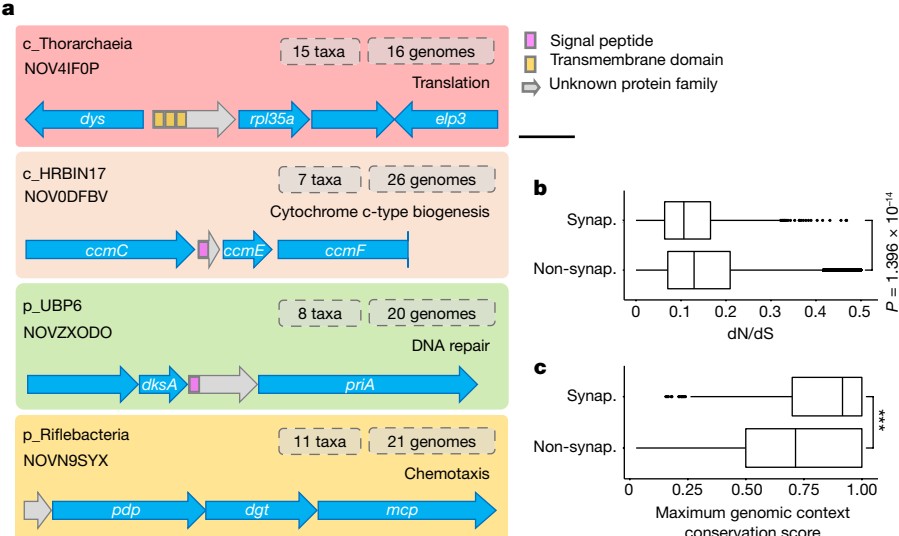

**Fig. 4 | FESNov synapomorphic gene families found at high-level taxonomic rank. a**, Schematic overview of four phylum- and class-level synapomorphic FESNov gene families located in conserved gene clusters involved in relevant cellular processes. Gene names correspond to: *dys* (K00809), deoxyhypusine synthase; *rpl35a* (K02917), large subunit ribosomal protein L35Ae; *elp3* (K07739), elongation protein 3 homolog; *ccmC* (K02195), haem exporter protein C; *ccmE* (K02197), cytochrome c-type biogenesis protein; *ccmF* (K02198), cytochrome c-type biogenesis protein; *dksA* (K06204), DnaK suppressor protein; *priA* (K04066), primosomal protein N'; *pdp* (K00756), pyrimidine-nucleoside phosphorylase; *dgt* (K01129), dGTPase; *mcp* (K03406), methyl-accepting chemotaxis protein. Scale bar, 500 base pairs. **b**, Comparison of dN/dS values between synapomorphic (synap., $n = 1,034$) and non-synapomorphic (non-synap., $n = 403,051$) FESNov gene families. **c**, Comparison of genomic context conservation between synap. ($n = 1,034$) and non-synap. ($n = 402,987$) FESNov gene families. **b**,**c**, Data are represented as boxplots in which the middle line is the median, the lower and upper hinges correspond to the first and third quartiles, the upper whisker extends from the hinge to the highest value no further than $1.5 \times$ interquartile range (IQR) from the hinge and the lower whisker extends from the hinge to the lowest value no further than $1.5 \times$ IQR of the hinge; data beyond the end of the whiskers are outlying points that are plotted individually. **b**,**c**, *P* values by two-sided Wilcoxon test are indicated (***$P < 1 \times 10^{-15}$).

Furthermore, the relative abundance profiles of FESNov families alone were sufficient to differentiate among environments and conditions, highlighting their ecological significance. Specifically, FESNov profiles across samples had a predictive power similar to those based on KEGG orthologous groups when comparing metagenomic samples from different environments using *t*-distributed stochastic neighbour-embedding (tSNE) multidimensional analysis (Extended Data Fig. 7).

### Discovery of new biomarkers

For assessment of the potential benefits of integration of the FESNov catalogue into biomarker discovery pipelines, we investigated its capacity to differentiate between metagenomic samples drawn from different conditions within the same environment, a common procedure in clinical studies. As a case study we revisited a recent microbiome investigation that assessed the predictive capability of functional modules for colorectal cancer (CRC) detection, based on a cohort of 575 human gut samples from five populations[46].

By inclusion of the FESNov catalogue in the analysis we were able to identify 69 novel gene families that were significantly overabundant in CRC samples compared with controls (false discovery rate (FDR) $q < 0.01$; Fig. 5 and Supplementary Table 13). Twenty-five of these overabundant gene families belong to the Acutalibacteraceae family, previously associated with CRC[47]. A further nine families are present in seven different species enriched in the gut of CRC patients[46]. Moreover, our functional predictions suggest that six of these families might be involved in motility, adhesion and invasion, processes potentially linked to colonization[48] (Extended Data Fig. 8).

The overall estimated accuracy of a CRC versus control diagnostic-oriented predictor, based solely on FESNov family abundance profiles, reached an area under the curve of 0.74. Predictors built on relative abundance matrices of both FESNov and known gene families (for example, KEGG orthologous groups) showed a slight but significant

increase of around 1.2% compared with those using known gene families only (Extended Data Fig. 9a). Overall, these results underscore the high utility of novel gene families for uncovering new functional biomarkers and their probable association with molecular mechanisms underpinning the observed differences.

### Discussion

With the genetic material of most of our planet's microbes remaining poorly studied, a wealth of genetic material with unknown potential for critical functional innovations remains largely overlooked and untapped by most metagenomic studies. In an effort to narrow this knowledge gap, we focused on identification of a highly curated set of previously uncharacterized microbial gene families of major evolutionary and functional significance. To this end we systematically addressed common problems such as sequencing artefacts, contaminations, gene fragments, false orphans, pseudogenes or sequences of insufficient quality or evidence—typical issues that continue to hamper the comparative analysis of the so-called microbial dark matter[1,49].

Despite nearly tripling the number of known prokaryotic orthologous groups described to date, our work has certain limitations that should be taken into account in future studies. For instance, the FESNov catalogue accounts for only a relatively small portion of all unknown sequences from the original dataset. Considering that many sequences and clusters were discarded due to quality filters and limited sampling depth, the catalogue probably represents only the proverbial tip of the iceberg, leaving many discoveries still to be made.

We also demonstrated that the FESNov catalogue can serve as a framework for the hypothesis-driven identification and experimental characterization of unknown but functionally important genes. However, functional associations based on genomic context analysis are usually broad and do not allow for precise predictions of new molecular

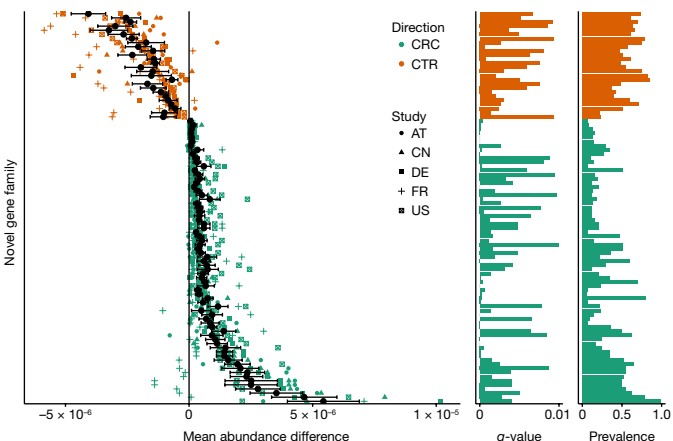

**Fig. 5 | FESNov gene families contribute to CRC predictive power and include biomarkers for the disease.** Difference in the relative abundance (colorectal cancer (CRC) versus control (CTR)) of FESNov gene families that are significantly over-represented ($n = 69$, shown in green) and under-represented ($n = 26$, shown in orange) in human gut samples ($n = 575$) from five different study populations (AT, Austria; CN, China; DE, Germany; FR, France; US, United States). Black points represent the mean relative abundance difference per FESNov family in the five studies; black bars indicate mean s.e. intervals. $P$ values by Wilcoxon test, adjusted ($q$-value) by the FDR method for multiple-hypothesis testing, are indicated in the $q$-value barplot. Prevalence (proportion of samples in which the FESNov family was detected) is indicated in the prevalence barplot.

functions. Whereas structural alignments can facilitate additional and more specific functional predictions, only 14% of FESNov gene families showed significant hits against known protein structures. This suggests that the majority of the unknown gene families encode for novel protein folds rather than undetectable distant homologues in the known functional landscape, posing new challenges for in silico functional predictions. Similarly, we identified many novel gene families that could be of great evolutionary significance for entire basal clades of uncultivated taxa; however, the taxonomic level at which these families were found to be synapomorphic may change with the discovery and sequencing of new species.

Finally we showed that, by integration of the FESNov catalogue into functional profiling tools such as the eggNOG-mapper tool, new ecological and clinical biomarkers can be detected. It is therefore crucial that the unique genetic repertoire of hitherto uncultivated species is regularly incorporated into reference databases and bioinformatic workflows, particularly as we advance in the exploration of understudied microbial ecosystems, micro-eukaryotic communities and low-abundance species.

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

# Methods

## Data collection

We retrieved medium- and high-quality MAGs, SAGs and reference genomes from five different studies: (1) 51,624 medium- and high-quality MAGs (at least 50% complete, contamination 5% or less) from a planetary multihabitat catalogue including 10,450 samples and covering diverse habitats (GEM)[3]; (2) 46,655 high-quality MAGs (at least 90% complete, contamination 5% or less) released by another multi-habitat catalogue spanning 13,174 samples and 14 habitats (GMGC)[2]; (3) 13,975 MAGs and SAGs and 17,935 reference genomes from the GTDB-r95 database[5]; (4) 26,975 medium- and high-quality MAGs (at least 50% complete, less than 10% contamination), 5,969 SAGs and 1,707 reference genomes obtained from ocean samples[4,50–52]; and (5) 4,644 medium- and high-quality reference human gut MAGs (at least 50% complete, less than 5% contamination) from the UHGG human gut catalogue[6]. Overall, the mean completeness and contamination of the collection of genomes were 86.15 and 1.13%, respectively (median 93 and 0.77%, respectively; Supplementary Table 14).

## Recalling ORFs

We noticed that some MAG collections contained genes predicted under an incorrect codon table (for instance, those from the Gracilibacteria or Mycoplasma lineages were predicted under the standard codon table), which could lead to incorrect conclusions[53]. We therefore recomputed ORF predictions for all MAGs in the GEM, GMGC and GTDB catalogues. For each MAG we used PROKKA[54], selecting the correct genetic table for each genome based on its taxonomic annotation. We further verified that the corrected Gracilibacteria and Mycoplasma ORFs were indeed longer than the original ones. We obtained 116,208,548, 110,913,525 and 106,052,079 genes for GEM, GMGC and GTDB, respectively. Following their combinion with the 56,637,438 and 10,002,521 genes in the oceanic and UHGG genomes, MAGs and SAGs we obtained a final catalogue of 399,814,111 genes.

## Taxonomic annotations

For normalizitaion of taxonomic annotations of all genomes in our collection, we reannotated all of them using GTDB-Tk v.1.6.0 (GTDB rev202 version)[55].

## Deep-homology clustering and phylogenetic tree reconstruction

For computation of gene family clusters we used MMseqs2 (ref. 19) with relaxed thresholds: minimum percentage of amino acids identity 30, $E$-value 0.001 or less and minimum sequence coverage 50%. The parameters used were --min-seq-id 0.3 -c 0.5 --cov-mode 1 --cluster-mode 2 -e 0.001. Clusters with fewer than three sequences were discarded. We computed multiple-sequence alignments for each gene family with Clustal Omega[56] using the translated version of the genes and subsequently reconstructed their phylogeny with FastTree2 (ref. 57) with default parameters. We calculated alignment statistics (mean identity, most unrelated pair, most distant sequence) on each gene family alignment using Alistat[58].

## Detection of protein clusters specifically from uncultivated taxa

For identification of genes/proteins without homologues in current genomic databases we mapped the members of each gene family cluster against (1) EggNOG v.5 (ref. 24) using eggNOG-mapper v.2 (ref. 39) and all protein sequences in each family as a query. Hits with an $E$-value below 0.001 were considered significant. (2) Pfam-A[59], using HMMER[60] hmmsearch against all protein sequences of each family. Hits with an $E$-value below $1 \times 10^{-5}$ were considered significant. (3) Pfam-B[23], using HMMER[60] hmmsearch searches against the representative protein sequence (longest sequence) of each family. Hits with an $E$-value below $1 \times 10^{-5}$ were considered significant. (4) Refseq[61], using DIAMOND[62] blastx (with the --sensitive flag) searches against the coding sequence (CDS) of all tmembers of the family. Hits with an $E$-value below 0.001 and query coverage above 50% were considered significant. All gene family clusters with a significant hit in any of the above databases were considered non-novel and discarded from the study. Additionally we mapped FESNov gene families against Eukprot[63], finding only 1,127 distant hits (Supplementary Table 15).

## Detection of spurious domains in gene families

To discard potential sequencing errors and pseudogenes we mapped our catalogue against the AntiFAM[26] database (HMMER search with the --cut_ga parameter, as recommended) and discarded families with $E$-value $1 \times 10^{-5}$ or less. We also discarded families with significant hits in the viral pVOG database[27] using HMMER, an $E$-value threshold of below $1 \times 10^{-5}$ and minimum coverage of 50%. Searches were performed at the amino acid level.

## Detection of conserved domains

We reconstructed protein-based, multiple-sequence alignments for each gene family with Clustal Omega. For each gene family the most conserved domain was considered the longest aligned region in which 80% of amino acid residues were not gaps. Gene families whose most conserved domain was shorter than 20 amino acid residues were discarded.

## Calculation of dN/dS

Multiple-sequence alignments from each gene family were back-translated into codon alignments to reconstruct phylogenetic trees using FastTree2 with default parameters. The entire workflow was executed using ETE3 (ref. 64) with options ete3 build --nt-switch-threshold 0.0 --noimg --clearall --nochecks -w clustalo_default-none-none-none --no-seq-rename. For calculation of selective pressure per family we ran HyPhy using the BUSTED model[65] with default parameters, codon-based nucleotide alignment and the phylogenetic tree generated previously, retrieving the dN/dS ratio under the full codon model. We discarded gene families with dN/dS values higher than 0.5.

## Detection of protein-coding families

We used back-translated, codon-based multiple-sequence alignments to run RNAcode[29]. Because the software calculates statistics on the longest sequence, we rearranged alignments so that the longest sequence was the first to appear. We ran RNAcode with default options and the --stop-early flag. Gene families yielding RNAcode $P$ values lower than 0.05 were considered coding and retained in our catalogue. Gene families without significant $P$ values were discarded from the study unless they were detected in metatranscriptomics datasets (see following section).

## Mapping to metatranscriptomic datasets

To obtain additional evidence of gene expression we mapped our gene families against (1) the TARA oceanic metatranscriptomic catalogue[21] and (2) 756 human gut metatranscriptomic samples[66]. For mapping the sequences of the novel families against TARA protein sequences we used DIAMOND blastp with the --sensitive flag. We considered any hit with an $E$-value below 0.001 and query coverage above 50% as significant. For mapping of reads from the human gut metatranscriptome samples against the sequences of novel families we used DIAMOND blastx sensitive mode. We considered any hit with an $E$-value below 0.001 as significant. Families were considered expressible if at least one member had a significant match against the metatranscriptomic catalogues.

## Orthology calling

All FESNov gene family clusters were analysed to determine whether they represented basal orthologous groups at the bacterial or archaeal

level or, by contrast, contained duplication events leading to several orthologous groups within the same family. To do so we rooted the phylogenetic tree of each family at midpoint and taxonomically annotated leaf nodes using GTDB v.202. Then, we used ETE[64] to identify duplication events on each phylogenetic tree with the 'get_descendant_evol_events' function and dated them according to their predicted common ancestry. Gene families without duplication events at basal taxonomic ranks were considered as representing orthologous groups at the bacterial or archaeal level.

## Comparison of gene family length distributions
We compared the distribution of protein lengths in eggNOG v.5 orthologous groups[24] and the small peptides catalogue described in ref. 17 with the length of the FESNov gene family clusters. The length of each gene family or orthologous group was set to the longest sequence within each cluster at the protein level.

## Sequence-based functional predictions
We ran SignalP-5.0 (ref. 67) with Gram-positive and Gram-negative modes on FESNov gene families. Genes predicted to encode for a signal peptide by either Gram-positive, Gram-negative or both were considered as secreted proteins. We also ran TMHMM[68] with default parameters to calculate transmembrane domains on the sequences. Similarly to the methods used in Sberro et al[17], a gene family was considered to encode for transmembrane/secreted proteins if at least 80% of its members were predicted to be so.

## Small peptide analyses
We considered FESNov gene families whose longest sequence was shorter than 50 residues to be small peptides. Our set of small peptides was mapped against those described by Sberro et al.[17] using DIAMOND (--sensitive flag), and considered as significant those with an $E$-value below 0.001 and coverage above 50%. We ran antimicrobial predictions on our small peptide FESNov gene families using Macrel[40].

## Detection of mobile elements
For the detection of families potentially included in plasmids we ran PlasFlow[69] with the --threshold 0.95 flag on all contigs from all MAGs, SAGs and genomes. For the detection of gene families with potential viral origin we ran Seeker[70] on all contigs from all MAGs, SAGs and genomes, using a threshold of 0.9 for considering a sequence as viral. For the results shown in Fig. 3 we considered gene families as mobile or viral if at least one member of the family was predicted to be so. The reported correlations were also found under more restrictive thresholds at which at least 30% of gene family members were predicted as mobile or viral sequences (Extended Data Fig. 6a).

## Ecological distribution of gene families
For expansion of the ecological profile of FESNov gene families we mapped the representative sequence of each family (longest) against 63,410 public metagenomic samples using DIAMOND (-sensitive flag). In total, 303,515 hits with an $E$-value below 0.001 and target coverage of at least 50% were considered significant. We grouped habitats into the following general groups: 'Human gut', 'Human oral', 'Marine', 'Human skin', 'Non-marine aquatic', 'Animal gut', 'Soil', 'Plant associated' and 'Anthropogenic'. Remaining samples were not included (that is, labelled as 'Other').

For building tSNE plots we mapped a representative sequence member of each family against the GMGC catalogue using DIAMOND (--sensitive flag). Hits with an $E$-value below 0.001 and target coverage of at least 50% were considered significant. For calculation of the abundance of each family on each sample we added the relative abundance of each gene family homologue. We restricted the tSNE analysis to families detected in more than 1,000 samples and to a maximum of 150 samples per habitat/human population. tSNE was computed with the python scikit-learn manifold package following logarithmic transformation of the data.

## Taxonomic breadth of gene families
For estimation of the taxonomic breadth of each FESNov gene family we calculated the last common ancestor (LCA) of their members using GTDB v.202 taxonomic predictions as the most lineage-specific annotation shared by all members of the family. To ensure that our LCA predictions were not artefacts caused by a small proportion of misannotated genes masking lineage-specific families to very basal levels we repeated the analysis, requiring the LCA lineage to be supported by 80% (Extended Data Fig. 6b) of the members of the family, thereby obtaining comparable patterns.

## Taxonomic distribution of FESNov gene families
To assess the distribution of novelty across the prokaryotic phylogeny we estimated the number of FESNov gene families observed per clade. Figure 3a shows the GTDB phylogenetic trees bac120_r202 and ar122_r202 collapsed to the order level (each tree leaf represents a taxonomic order). For representation of the percentage of uncultivated genomes per branch in Fig. 3a, we divided the number of uncultivated genomes per lineage by the total number of genomes under that lineage. The final tree image was generated using Interactive Tree Of Life[71].

## Synapomorphic gene families
For identification of synapomorphic gene families at different taxonomic levels we calculated the clade specificity and coverage of each gene family across all GTDB v.202 lineages. For each gene family and clade, coverage was calculated as the number of genomes containing a specific gene family over the total number of genomes under the target clade. Specificity was estimated as the percentage of protein members within a family that belonged to the target clade.

Under such criteria, very few synapomorphic gene family clusters would be expected at basal taxonomic levels, typically indicating lineage-specific functional traits. Preceding our analysis, and to verify that our detection method was correct, we computed synapomorphic gene families out of the complete set of clusters obtained in the first step of the pipeline using a cutoff of 70% coverage and 90% specificity. As expected, we detected a very low number (34,370) of synapomorphic clusters at high-level taxonomic ranks (phylum, class or order; Supplementary Table 16), which included both known and unknown gene families. Among functionally known synapomorphies we found previously described examples of synapomorphies, such as the photosystem P840 reaction centre protein (*pscD*) and the chlorosome envelope protein C (*csmC*), which are known to be synapomorphic at the *Chlorobium* bacterial class[72,73]. Similarly, photosystem I subunits *psaD* and *psaF*, photosystem II oxygen-evolving enhancer protein *psbO* and the NHD1 complex subunit (*ndhN*) were also detected as synapomorphic for the entire Cyanobacteria phylum. In addition, we validated our method for location of synapomorphic gene families by calculation of coverage and specificity within each lineage for each KEGG orthologous group and compared the conservation values obtained with those in a recent study[74], finding a correlation coefficient of 0.71 ($P < 1 \times 10^{-15}$).

To obtain the final set of synapomorphic gene families in the FESNov catalogue (see above), and to focus only on the most prominent examples, we used the same validated method but with stricter specificity and coverage thresholds, requiring also a minimum number of members per family. Thus we considered FESNov gene families as synapomorphic if they contained more than ten members (that is, protein sequences) and had a coverage higher than 90% and specificity of 100% for a given lineage. We provide the coverage and specificity values of all families from every lineage in the downloads section of https://novelfams.cgmlab.org/. Moreover, to further ensure that our final set of synapomorphic predictions is highly specific, we mapped

them back against the whole catalogue using a more sensitive mapping strategy based on HMMER searches[60], excluding families with distant hits to genomes that might compromise our specificity and coverage thresholds.

For comparison of the degree of genomic context conservation between synapomorphic and non-synapomorphic FESNov gene families shown in Fig. 4c, we computed the syntenic conservation of adjacent neighbour genes (+1 and −1 positions).

## Functional predictions

To infer the functional annotation of the neighbouring genes of FESNov gene families we ran eggNOG-mapper v.2 (ref. 39) with default parameters on the 400 million proteins in our catalogue of MAGs, SAGs and reference genomes. We also mapped them against the CARD[75] database using DIAMOND blastp, with E-value and coverage threshold of 0.001 and over 50%, respectively. We next built a database with all the genomic locations of genes in their respective contigs and the annotated function of every gene.

We then established an association between a KEGG pathway (P) and a FESNov family if 30, 50, 80 or 90% of FESNov neighbour genes in a genomic window of plus or minus three genes of all FESNov members were annotated to pathway P (synteny conservation), calculating also the proportion of neighbour genes annotated to pathway P over the total number of neighbours within the same window (functional relatedness) and measuring the frequency of neighbour genes annotated to pathway P that are (1) on the same orientation as FESNov family members (strand conservation) and (2) with no intergenic regions longer than 100 nucleotides (distance conservation).

To determine an approximate confidence level for all such associations we computed the same calculations for 108,823 gene family clusters with known and consistent KEGG pathway annotations. We then estimated the minimum synteny conservation, functional relatedness, strand conservation and distance conservation necessary to successfully recover the function (KEGG pathway) of at least 50 or 90% of all families (confidence scores) assigned to that KEGG pathway. The implementation of this method and benchmark is available at https://github.com/AlvaroRodriguezDelRio/nov-fams-pipeline.

Using a confidence score of 90%, we were able to functionally annotate 1.2% of the FESNov gene families detected in host-associated samples but only 0.9% of pure free-environmental samples. For computation of FESNov gene families in resistance islands we identified the number of positions in which the family had CARD genes within https://github.com/arpcard/aro with different evolutionary conservation values. To generate Fig. 2a we joined the bacterial and archaeal GTDB phylogenetic trees, collapsed to the order level, using the ETE toolkit software. We then represented the presence/absence matrix of novel gene family predictions associated with KEGG pathways with confidence larger than 0.9 across the different taxonomic orders. For readability, orders and KEGG pathways with no predictions are not shown in the figure.

We also mapped the representative sequence of each novel family against the collection of functionally annotated genes in Miller et al.[33] with DIAMOND blastp (sensitive flag). Hits with an E-value below 0.001 and coverage above 50% on the FESNov family were considered significant. We then compared our predictions with theirs, and used them to extend our functional predictions. Both approaches reached 81% agreement for functional predictions tagged as high confidence. Mappings against Miller et al.[33] provided genome context-based functional predictions to 22,898 FESNov gene families.

We also mapped FESNov gene family sequences against Fitness Browser[38] genes (https://fit.genomics.lbl.gov/cgi_data/aaseqs) using DIAMOND blastp with maximum E-value and minimum coverage thresholds of 0.001 and 50%, respectively. We considered hits with strong fitness changes (t-score above 4 or below −4; see original publication) to be potentially associated with certain conditions.

For location of FESNov gene families in biosynthetic clusters we screened the Antismash[76] predictions presented in Paoli et al.[4], whose genes are included in this catalogue.

## Structural predictions

As an initial attempt, we first queried all FESNov gene families against a large database of protein structural models of environmental sequences[77]. To do so we mapped the longest member of each novel gene family against the sequences of High Confidence MGnify30 (ref. 78) using DIAMOND with E-value below 0.001 and coverage above 50% significance thresholds. However, we found matches for only about one-third (33.6%) of our FESNov families.

Given the low number of matches obtained, we decided to run de novo structural predictions for all FESNov families by means of ColabFold[35] using the representative sequence of each family as input. In total we computed structural predictions for 389,638 FESNov gene families. For the remainder, ColabFold was not able to finish computations. For the FESNov gene families for which we could obtain structures, we ran Foldseek[79] against PDB[36] (https://rcsb.org) and Uniprot[37] (-e E-value, 0.001 -s sensitivity, 9.5 options). Structures were considered reliable if the PLDDT score was at least 70, as suggested in the literature[80].

## Motility assays

The coding sequence of a representative member of the NOV3845Y gene family (nov3845y) was manually codon optimized for expression in E. coli W3110 strain and subsequently cloned into pBlueScript II SK(+) using BamHI and KpnI sites. This plasmid contains an isopropyl-β-D-thiogalactoside (IPTG) inducible lac promoter. A Shine–Dalgarno consensus sequence (AGGAGG) was inserted at the beginning of the sequence to ensure ribosome binding and translation of the transcript. The plasmid was synthesized by BioCat. The representative sequence from the NOV3845Y gene family was: GTGGAC GATCTGTTCATAATAGCCAGATGGAATGCGCAATATTTTGCCTGTGCGG AAGCAGACACGAAAGAAATTCTTGGTGGGCTGAATCAGGGCAGTCAA AGTCGAGGTGAAGAGGATGCCAGAGACGTTCAAAAGCATCATTTGTTT GCAGATCATGCGCTGGTGGAGTGCAACTTCGGTATCCTTCTAAACGAC AATCAAATAGTGCCAAGTGACGATATTCCATCGCGGGTTTCGGCGACA TTTTTGCCGCTGGACCCATATGGCGGTACGATCCTGAAAGACATCACC GGTATTATCCAGATTGACGAGGGTACATTCGCCCTTGTGCTGGCGCCT TCAAAAAACGATGCCAGAGGGGGATCGCTTGGAACAGGCCTTGATTAA, which was codon optimized to ATGGACGATCTGTTCATCATCGCCC GCTGGAATGCGCAGTATTTCGCCTGCGCGGAAGCAGACACGAAAGAAA TCCTTGGTGGGCTGAATCAGGGCAGCCAGAGCCGCGGTGAAGAGGA TGCCCGCGACGTGCAGAAGCATCATCTGTTCGCAGATCATGCGCTGGT GGAGTGCAACTTCGGTATCCTTCTGAACGACAATCAGATCGTGCCGA GCGACGATATCCCGTCGCGGGGTGTCGGCGACCTTCCTGCCGCTGGAC CCGTATGGCGGTACGATCCTGAAAGACATCACCGGTATCATCCAGATC GACGAGGGTACGTTCGCCCTTGTGCTGGCGCCGAGCAAAAACGATGC CCGCGGGGGATCGCTTGGAACCGGCCTTGATTAA.

A plasmid containing the selected sequence nov3845y and the empty vector were transferred into E. coli W3110 strain by electroporation. Strains that successfully incorporated plasmids were selected for based on their resistance to ampicillin and checked by PCR with M13 universal primers.

For swimming motility experiments, an E. coli strain overexpressing NOV3845Y was first streaked on solid lysogeny broth medium supplemented with ampicillin 100 μg ml⁻¹ (Ap100). To prepare the swimming plates, 50 ml of semisolid lysogeny broth Ap100 containing 0.3% agar and IPTG at concentrations of 0 μM, 10 μM, 100 μM and 1 mM was poured onto squared plates and left open to dry for 10 min in the biosafety laminar flow cabin. Isolated colonies were then inoculated with a toothpick by perforation of the medium, avoiding reaching the bottom of the plate. Swimming area was quantified for each inoculated colony by measuring the diameter of the

bacterial ring following overnight growth (roughly 16 h) in a humid environment.

### Antimicrobial activity assay

The candidate antimicrobial peptide NOVOQR9B (sequence: MQT-VNEPNVTGATPRGGCFVKTGCGKYKGSCTIHLA) was chemically synthesized by BioCat. For analysis of the putative antibacterial activity of the peptide, three Gram-positive bacterial strains (*P. polymyxa*, *B. subtilis* and *Lactobacillus* sp.) and two Gram-negative bacterial strains (*E. coli* K12 and *Ps. putida* KT2440) were selected. Bacterial strains were grown overnight at 28 °C in lysogeny broth agar plates. Cells were then scraped from the plates, resuspended in 1/3-diluted NB medium (1 g of yeast extract, 2 g of beef extract, 5 g of NaCl and 5 g of Bacto Peptone (per litre)) and centrifuged at 5,000$g$ for 5 min. The resulting pellet was washed twice with 1/3-diluted NB medium and adjusted to an optical density at 600 nm ($OD_{600}$) of 1.

Antimicrobial activity assays were performed in flat-bottomed, 96-well plates. The NOVOQR9B peptide was initially dissolved in 1/3-diluted NB medium to a concentration of 2 mM and then serially diluted in the same medium to reach final concentrations ranging from 0.137 to 300 μM. Bacterial suspensions were then added to wells containing the peptide (final $OD_{600}$ of 0.1). Plates were incubated at 28 °C with shaking at 200 rpm, and the $OD_{600}$ values of each well were measured following culture for 24 h using a SPECTROstar Nano microplate reader (BMG LABTECH). The assay was carried out in triplicate.

### Functional profiling and biomarker discovery

We calculated the relative abundance of FESNov gene families on 575 human gut samples from a clinical study involving 285 CRC patients and 290 controls from five different populations[46]. For this we mapped the longest representative of each FESNov gene family against this dataset with DIAMOND blastp and the --sensitive flag. Hits with *E*-value below 0.001 and coverage on the novel family sequence of over 50% were considered significant. For estimation of the relative abundance of a FESNov family in a given sample we summed the relative abundance of all significant hits in that sample. We computed KEGG ortholog abundance by summing the relative abundances of all genes annotated with a particular KEGG ortholog; KEGG ortholog annotations were obtained with eggNOG-mapper v.2.

Differentially abundant FESNov gene families between CRC and control samples were estimated, per gene family, using a blocked Wilcoxon test as implemented in the 'coin' R package, correcting study effects by blocking for 'study' and 'colonoscopy'. We corrected *P* values by the FDR method to adjust for multiple-hypothesis testing to obtain *q*-values. A FESNov gene family was considered significantly over- or underabundant in CRC samples compared with control samples if *q*-values were below 0.01.

For comparison of the predictive power of the relative abundance profiles built on FESNov gene families versus KEGG orthologies, and the combination of the two (FESNov + KEGG orthologies), we ran logistic regression models as implemented in the 'gmlnet' R package (Extended Data Fig. 9a) and random forest algorithms as implemented in the 'caret' R package (Extended Data Fig. 9b). We ran the models independently for the relative abundance tables of FESNov, KEGG orthologies and FESNov + KEGG orthologies, training the models with 70% of the samples and testing the prediction performance on the remaining 30%. For the random forest analysis, 1,000 random gene families were selected. Following iteration of the analysis on different train and test data, we compared area under the curve values obtained when considering the relative abundance of FESNov gene families, KEGG orthologies and the combination of the two.

### Reporting summary

Further information on research design is available in the Nature Portfolio Reporting Summary linked to this article.

### Data availability

All genomic data used in this study were downloaded from public sources as follows: UGHH MAGs from https://www.ebi.ac.uk/ena/browser/view/PRJEB33885, Ocean MAGs and SAGs from https://www.ebi.ac.uk/ena/browser/view/PRJEB45951 and https://microbiomics.io/ocean/, GMGC MAGs from https://gmgc.embl.de, GEM MAGs from https://genome.jgi.doe.gov/GEMs and GTDB reference genomes and MAGs from https://data.gtdb.ecogenomic.org/releases/release95/95.0. All the results derived from this study, including FESNov gene family fasta files, phylogenetic trees and alignments, FESNov gene family statistics and evolutionary information, mobile element detections, taxonomic annotations, functional prediction summaries and protein structure predictions, are available at https://zenodo.org/doi/10.5281/zenodo.10219528. Computer-generated structural models are also available at https://modelarchive.org/doi/10.5452/ma-fesnov. In addition, large intermediate files from some analyses are provided at https://novelfams.cgmlab.org/downloads/, including: all standardized genomes, MAGs and SAGs downloaded from public sources, consolidated FASTA files with predicted genes and proteins, functional annotations of all proteins by eggNOG-mapper v.2.1 and raw clustering results. Source data are provided with this paper.

### Code availability

Custom code was developed for some of the analyses included in the manuscript and is available at https://zenodo.org/doi/10.5281/zenodo.10209798. In addition, this code is also available as GitHub repositories at https://github.com/AlvaroRodriguezDelRio/NovFamilies and https://github.com/AlvaroRodriguezDelRio/nov-fams-pipeline, including a description on how to apply different-quality filters, predict pathways based on genomic context conservation analysis and calculate taxonomic conservation values. Implementation of the FESNov-based eggNOG-mapper functional profiler is available at https://github.com/eggnogdb/eggnog-mapper and can be used in the software from v.2.1.12 onwards and via the online service at http://eggnog-mapper.embl.de/.

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

**Acknowledgements** We thank Y. Duan and C. Dias Santos Júnior (Fudan University) for their help with the habitat annotation of metagenomes. We thank G. Zeller and J. Wirbel for critical discussion on the functional profiling of CRC samples. We thank X. Jiang, L. Rubio, K. Patil, I. Roux and S. Blasche for their support and discussion on the experimental validations of novel functions. Editorial assistance, in the form of language editing and correction, was provided by XpertScientific Editing and Consulting Services. This project has received funding from the National Programme for Fostering Excellence in Scientific and Technical Research (grant no. PGC2018-098073-A-I00 MCIU/AEI/FEDER, UE) and, partially, by MCIN/AEI/10.13039/501100011033/ and FEDER Una manera de hacer Europa (grant no. PID2021-127210NB-I00). A.R.d.R. was supported by a fellowship from la Caixa Foundation (ID 100010434, fellowship code no. LCF/BQ/DI18/11660009), cofunded by the European Union's Horizon 2020 research and innovation programme under Marie Skłodowska-Curie grant agreement no. 713673. C.P.C., S.S.-H. and Z.D. acknowledge support by Severo Ochoa Centres of Excellence Programme from the State Research Agency of Spain (grant nos. SEV-2016-0672 (2017–2021) and CEX2020-000999-S). J.B. acknowledges support by a grant from the Chan Zuckerberg Initiative DAF, an advised fund of Silicon Valley Community Foundation (no. 2020-218584). A.H.-P. was supported by Research Technical Support Staff Aid (no. PTA2019-017593-I/AEI/10.13039/501100011033). M.M.-P., J.J.R.-H. and E.L.-S. acknowledge support from Ministerio de Ciencia e Innovación, MCIN/AEI/10.13039/501100011033 (grant no. PID2021-125673OB-I00). S.S. acknowledges support from the Swiss National Science Foundation project (grant no. 205321_184955) and NCCR Microbiomes (no. 51NF40_180575), and thanks the staff at ETH Zurich IT Services and HPC facilities.

**Author contributions** J.H.-C. conceived the project, designed and supervised experiments and performed part of the computational analyses. A.R.d.R. designed experiments, collected data and performed all computational analyses, benchmarks and statistical tests. J.G.-L. designed and performed part of the genomic context analyses and curated functional predictions for experimental validations. C.P.C. implemented the eggNOG-mapper functional annotation module for the FESNov catalogue. J.B. implemented part of the online resource. Z.D. computed functional annotations based on protein structures and helped with online resource implementation. A.H.-P. participated in the clustering of gene families. M.M.-P., S.S.-H., J.J.R.-H. and E.L.-S. designed and performed experimental validations of gene families NOV3845Y and NOVOAR9B. H.-J.R., L.P. and S.S. helped establish the genomic dataset, computed most of the protein structure predictions and contributed mappings to biosynthetic clusters from marine metagenomes. T.S.B.S., P.B. and L.P.C. contributed ecological mappings and developed the habitat ontology used to compute ecological distributions. J.H.-C., A.R.d.R. and J.G.-L. wrote the paper and prepared all figures with feedback from all authors. All of the authors read and approved the final manuscript.

**Competing interests** The authors declare no competing interests.

**Additional information**
**Correspondence and requests for materials** should be addressed to Jaime Huerta-Cepas.

**a**

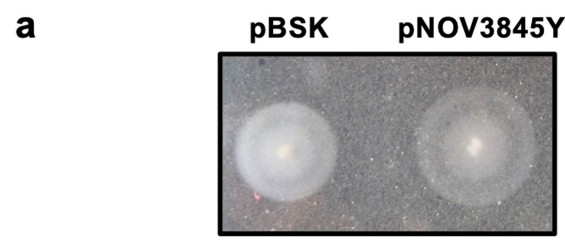

**b**

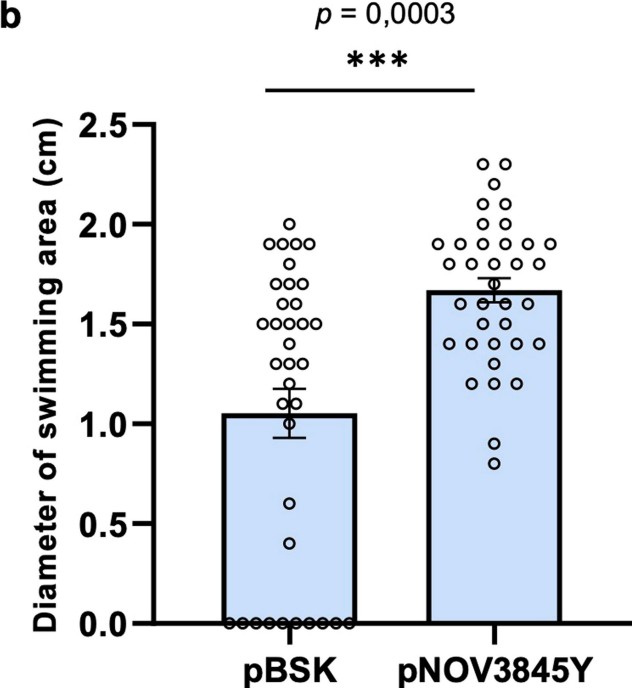

**Extended Data Fig. 1 | Swimming chemotaxis assay of *Escherichia coli W3110* strain expressing NOV3845Y.** (a) Representative image of a swimming chemotaxis assay of *Escherichia coli W3110* strains containing either *pBSK* or *pNOV3845Y*. (b) Quantification of the swimming area (measured as diameter in cm) of *Escherichia coli W3110* strains harboring *pBSK* (n = 36, with 3 biological replicates, each containing 12 technical replicates) and *pNOV3845Y* (n = 36, with 3 biological replicates, each containing 12 technical replicates). Data are represented as mean values ± SEM. p-value by two-tailed Mann-Whitney U test is indicated.

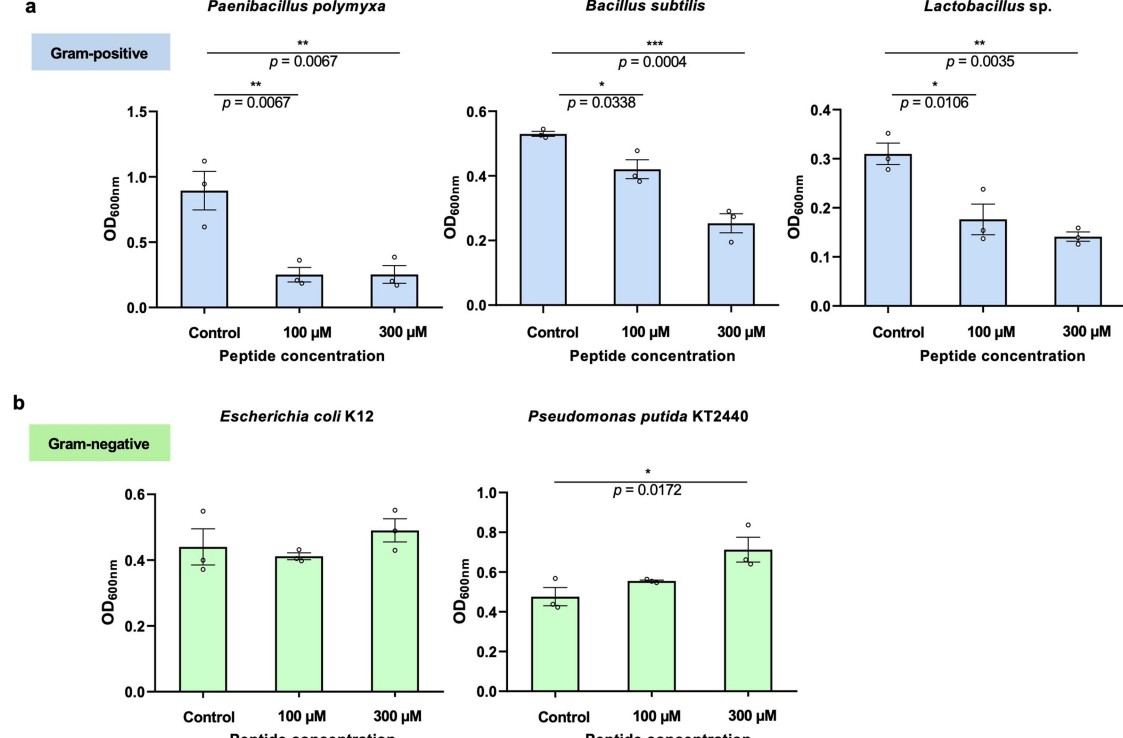

**Extended Data Fig. 2 | Antimicrobial activity of NOVOQR9B peptide.**
Growth of 3 Gram-positive (a) and 2 Gram-negative (b) bacterial strains in 1/3 diluted NB medium with peptide concentration as indicated. Control bars represent bacterial culture growth in the absence of the peptide. Only the results for the two highest peptide concentrations are shown. Data from 3 technical replicates for all bacterial strains tested are represented as mean values ± SEM. p-values by One-way ANOVA test are indicated (*p < 0.05; **p < 0.01; ***p < 0.001).

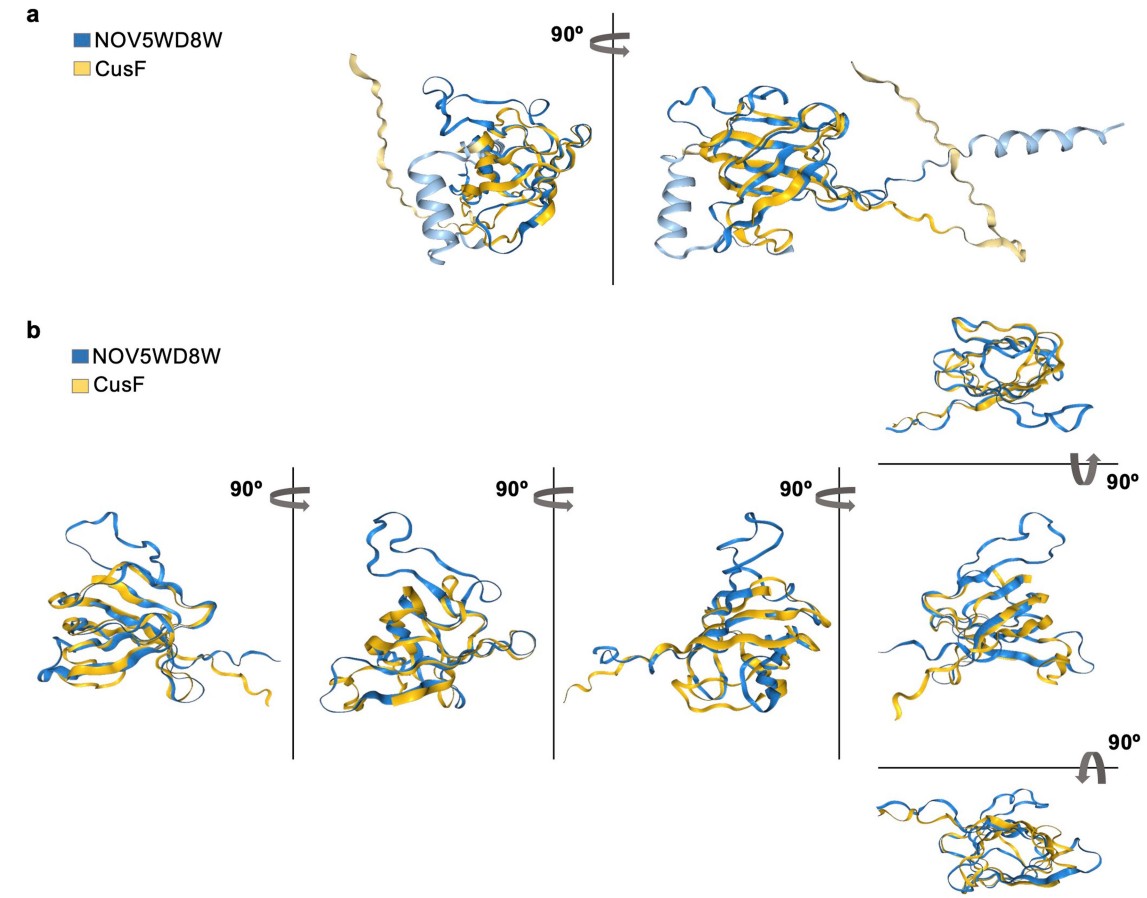

**Extended Data Fig. 3 | Structural similarity of NOV5WD8W with the copper metallochaperone CusF.** (a) Superposition of the structures of FESNov protein sequence NOV5WD8W (blue) and *CusF* (yellow) from *Escherichia coli*.

(b) Zoom-in on the high-confidence structural homology domain in superposed structure of the FESNov protein sequence NOV5WD8W (blue) and *CusF* (yellow) from *Escherichia coli*.

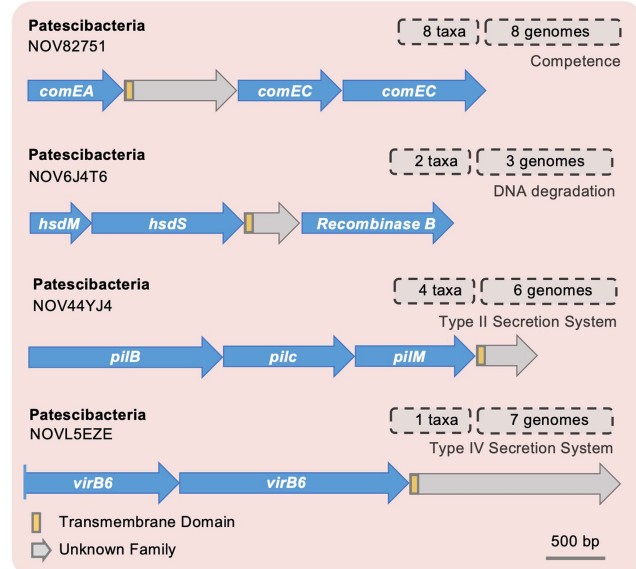

**Extended Data Fig. 4 | Schematic representation of the genomic context of four transmembrane FESNov gene families in Patescibacteria genomes.** Gene names correspond to *comEA* (K02237): competence protein *ComEA*, *comEC* (K02238) competence protein *ComEC*; *hsdM* (K03427): type I restriction enzyme M protein, *hsdS* (K01154): type I restriction enzyme, S subunit, recombinase B; *pilB* (K02652): type IV pilus assembly protein PilB, *pilC* (K02653): type IV pilus assembly protein PilC, *pilM* (K02662): type IV pilus assembly protein PilM; *virB6* (K03201: type IV secretion system protein VirB6).

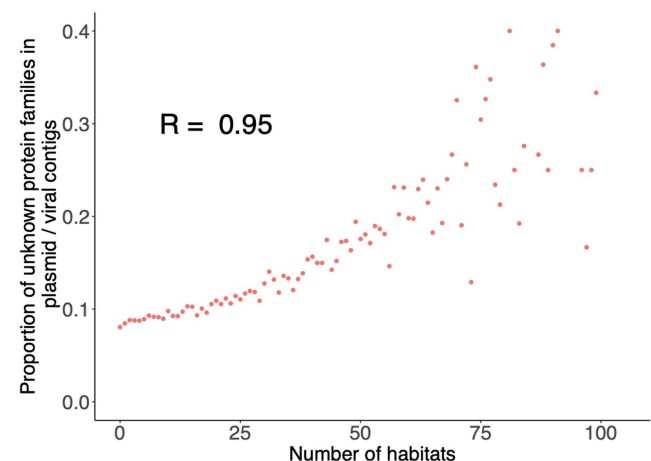

**Extended Data Fig. 5 | Correlation between FESNov gene families mobility and ecological dispersion.** Proportion of FESNov gene families linked to plasmids or viral contigs in relation to the number of habitats they were detected in. R indicates the Spearman correlation coefficient.

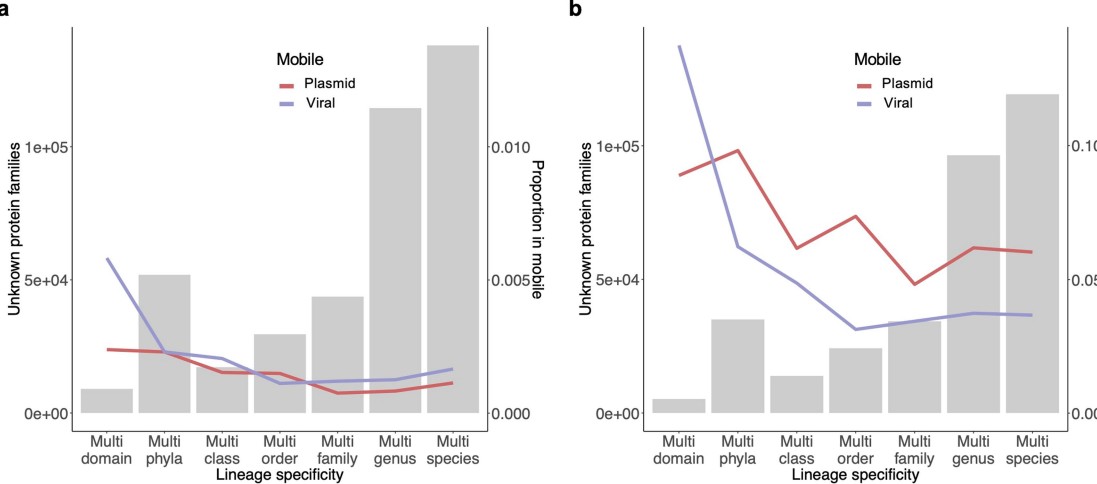

**Extended Data Fig. 6 | Number of FESNov gene families confined to each taxonomic rank.** Number of FESNov gene families confined to each taxonomic rank attending to the last common ancestor (LCA) of their family members. The lines indicate the proportion of gene families detected in plasmids (red) or viral (blue) contigs. This figure is equivalent to Fig. 3c with the following differences **(a)** requiring 30% of the members to be present in plasmids/viral contigs for considering the family as mobile and **(b)** calculating LCAs as the most basal taxonomic group gathering 80% of the members of the family.

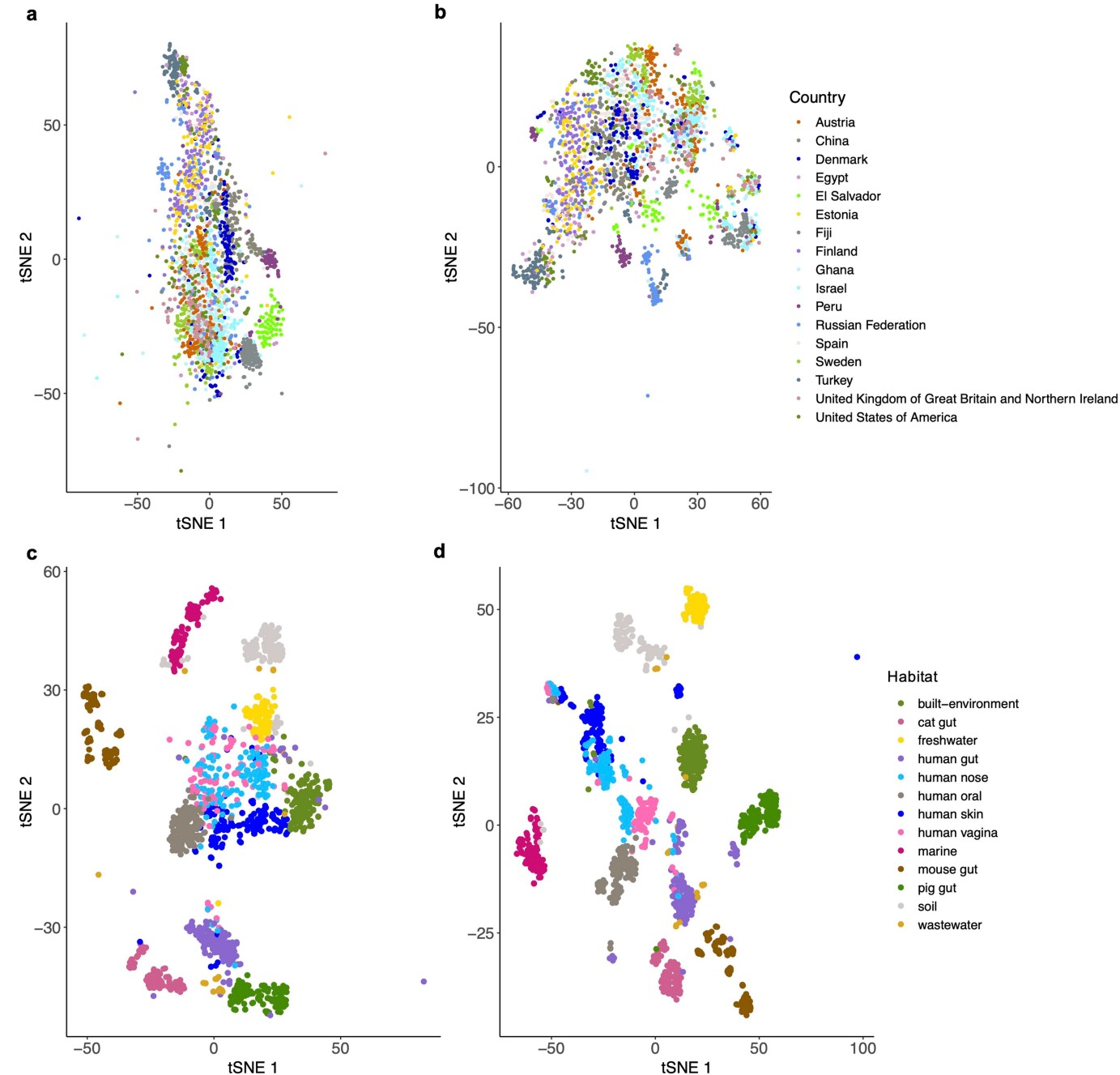

**Extended Data Fig. 7 | Separation of habitats and human gut populations with FESNov families and KO relative abundances.** tSNE representation of the log-transformed abundances of FESNov gene families and KOs detected in more than 1,000 samples within the GMGC catalog. To correct for differences in the number of samples available per habitat and human gut sample available per country, we included a maximum of 150 random samples per habitat/country. (a) tSNE representation of FESNov gene families per country; (b) tSNE representation of KOs per country; (c) tSNE representation of FESNov gene families per habitat; (d) tSNE representation of KOs per habitat.

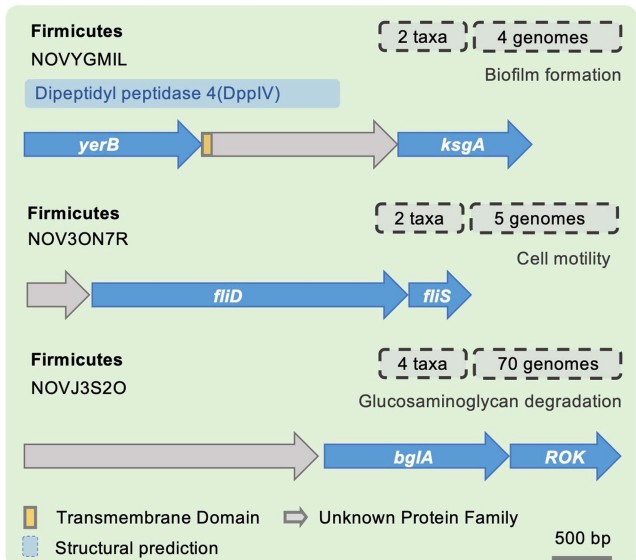

**Extended Data Fig. 8 | Examples of the genomic context of FESNov families over-abundant in CRC samples.** Schematic overview of the genomic context of NOVYGMIL, NOV3ON7R, and NOVJ3S2O; FESNov gene families significantly more abundant in CRC samples with conserved genomic context related to cell adhesion, motility, and cell invasion. Gene names correspond to: *yerB* (COG1470), cell adhesion and biofilm formation; *ksgA* (K02528), 16S rRNA (adenine1518-N6)-dimethyltransferase; *FliD* (K02407), flagellar hook-associated protein, *fliS* (K02422), flagellar secretion chaperone; *bglA* (K01223), 6-phospho-beta-glucosidase, ROK, sugar kinase protein (regulatory protein).

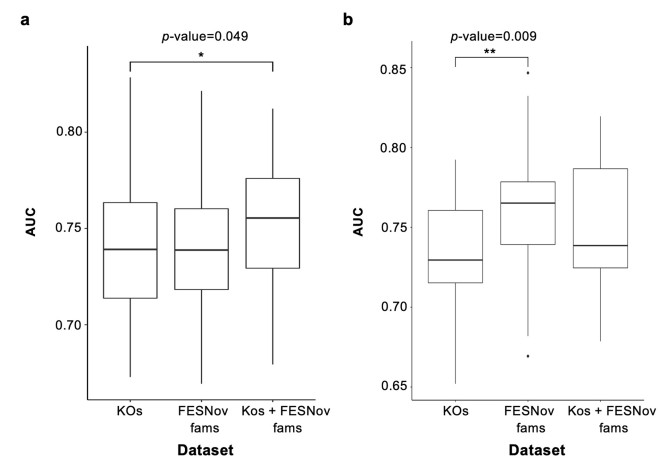

**Extended Data Fig. 9 | Performance of predictors built upon the relative abundance matrices of both FESNov gene families and KEGG Orthologs (KOs) families. (a)** Area Under the Curve (AUC) values obtained from 50 iterations of a logistic regression model built on KEGG Orthologs (KOs), FESNov gene families, and the combination of both **(b)** AUC values obtained from 30 iterations of a random forest model built on KOs, FESNov gene families and the combination of both, Data are represented as boxplots where the middle line is the median, the lower and upper hinges correspond to the first and third quartiles, the upper whisker extends from the hinge to the largest value no further than 1.5 × IQR from the hinge (where IQR is the interquartile range) and the lower whisker extends from the hinge to the smallest value at most 1.5 × IQR of the hinge, while data beyond the end of the whiskers are outlying points that are plotted individually. *p*-values by two-sided Wilcoxon-test are indicated.

# Reporting Summary

## Statistics

For all statistical analyses, confirm that the following items are present in the figure legend, table legend, main text, or Methods section.

| n/a | Confirmed | |
|---|---|---|
| ☐ | ☒ | The exact sample size (*n*) for each experimental group/condition, given as a discrete number and unit of measurement |
| ☐ | ☒ | A statement on whether measurements were taken from distinct samples or whether the same sample was measured repeatedly |
| ☐ | ☒ | The statistical test(s) used AND whether they are one- or two-sided<br>*Only common tests should be described solely by name; describe more complex techniques in the Methods section.* |
| ☒ | ☐ | A description of all covariates tested |
| ☐ | ☒ | A description of any assumptions or corrections, such as tests of normality and adjustment for multiple comparisons |
| ☐ | ☒ | A full description of the statistical parameters including central tendency (e.g. means) or other basic estimates (e.g. regression coefficient) AND variation (e.g. standard deviation) or associated estimates of uncertainty (e.g. confidence intervals) |
| ☐ | ☒ | For null hypothesis testing, the test statistic (e.g. *F*, *t*, *r*) with confidence intervals, effect sizes, degrees of freedom and *P* value noted<br>*Give P values as exact values whenever suitable.* |
| ☒ | ☐ | For Bayesian analysis, information on the choice of priors and Markov chain Monte Carlo settings |
| ☒ | ☐ | For hierarchical and complex designs, identification of the appropriate level for tests and full reporting of outcomes |
| ☒ | ☐ | Estimates of effect sizes (e.g. Cohen's *d*, Pearson's *r*), indicating how they were calculated |

*Our web collection on statistics for biologists contains articles on many of the points above.*

## Software and code

Policy information about availability of computer code

| | |
|---|---|
| Data collection | No specific software was used to collect data. Original datasets were manually downloaded from their public sources. |
| Data analysis | MMseqs2, version,  https://github.com/soedinglab/MMseqs2<br>diamond, version v0.9.32.133,  https://github.com/bbuchfink/diamond<br>HMMER, version 3.2,  http://hmmer.org/<br>clustalO, version 1.2.4: http://www.clustal.org/omega/<br>FastTree, version 2.1, : http://www.microbesonline.org/fasttree/<br>ETE3, version 3.1.2, : http://etetoolkit.org/download/<br>HyPhy, version  2.5.14, : https://www.hyphy.org/<br>RNAcode, version 0.3: https://github.com/ViennaRNA/RNAcode<br>mongoDB, version 3, : https://www.mongodb.com/<br>GTDB-Tk, version v1.6.0,: https://github.com/Ecogenomics/GTDBTk<br>Macrel, version  0.6.1, https://github.com/BigDataBiology/macrel<br>Seeker,  https://github.com/gussow/seeker<br>Plasflow, version 1.1, https://github.com/smaegol/PlasFlow<br>SignalP, version 5.0, https://services.healthtech.dtu.dk/services/SignalP-5.0/<br>TMHMM version 2.0. https://services.healthtech.dtu.dk/services/TMHMM-2.0/<br>Alistat, http://www.csb.yale.edu/userguides/seq/hmmer/docs/node27.html |

For manuscripts utilizing custom algorithms or software that are central to the research but not yet described in published literature, software must be made available to editors and reviewers. We strongly encourage code deposition in a community repository (e.g. GitHub). See the Nature Portfolio guidelines for submitting code & software for further information.

## Data

Policy information about [availability of data](availability of data)

All manuscripts must include a [data availability statement](data availability statement). This statement should provide the following information, where applicable:

- Accession codes, unique identifiers, or web links for publicly available datasets
- A description of any restrictions on data availability
- For clinical datasets or third party data, please ensure that the statement adheres to our [policy](policy)

All genomic data used in this study were downloaded from public sources as follows:
UGHH MAGs from https://www.ebi.ac.uk/ena/browser/view/PRJEB33885, Ocean MAGs and SAGs from https://www.ebi.ac.uk/ena/browser/view/PRJEB45951 and https://microbiomics.io/ocean/, GMGC MAGs from https://gmgc.embl.de, GEM MAGs from https://genome.jgi.doe.gov/GEMs, and GTDB reference genomes and MAGs from https://data.gtdb.ecogenomic.org/releases/release95/95.0. All the derived results from this study, including FESNov gene families fasta files, phylogenetic trees and alignments, per FESNov gene family statistics and evolutionary information, mobile element detections, taxonomic annotations, functional prediction summaries and protein structure predictions are available at https://zenodo.org/doi/10.5281/zenodo.10219528. All genomic data used in this study were downloaded from public sources as follows:
UGHH MAGs from https://www.ebi.ac.uk/ena/browser/view/PRJEB33885, Ocean MAGs and SAGs from https://www.ebi.ac.uk/ena/browser/view/PRJEB45951 and https://microbiomics.io/ocean/, GMGC MAGs from https://gmgc.embl.de, GEM MAGs from https://genome.jgi.doe.gov/GEMs, and GTDB reference genomes and MAGs from https://data.gtdb.ecogenomic.org/releases/release95/95.0. All the derived results from this study, including FESNov gene families fasta files, phylogenetic trees and alignments, per FESNov gene family statistics and evolutionary information, mobile element detections, taxonomic annotations, functional prediction summaries and protein structure predictions are available at https://zenodo.org/doi/10.5281/zenodo.10219528. Computer generated structural models are also available at https://modelarchive.org/doi/10.5452/ma-fesnov. In addition, large intermediate files from some analyses are provided at https://novelfams.cgmlab.org/downloads/, including: all the standardized genomes, MAGs and SAGs downloaded from public sources, consolidated FASTA files with predicted genes and proteins, functional annotations of all proteins by eggNOG-mapper v2.1 and raw clustering results.
In addition, large intermediate files from some analyses are provided at https://novelfams.cgmlab.org/downloads/, including: all the standardized genomes, MAGs and SAGs downloaded from public sources, consolidated FASTA files with predicted genes and proteins, functional annotations of all proteins by eggNOG-mapper v2.1 and raw clustering results.

## Research involving human participants, their data, or biological material

Policy information about studies with [human participants or human data](human participants or human data). See also policy information about [sex, gender (identity/presentation), and sexual orientation](sex, gender (identity/presentation), and sexual orientation) and [race, ethnicity and racism](race, ethnicity and racism).

| | |
|---|---|
| Reporting on sex and gender | N/A |
| Reporting on race, ethnicity, or other socially relevant groupings | N/A |
| Population characteristics | N/A |
| Recruitment | N/A |
| Ethics oversight | N/A |

Note that full information on the approval of the study protocol must also be provided in the manuscript.

# Field-specific reporting

Please select the one below that is the best fit for your research. If you are not sure, read the appropriate sections before making your selection.

☐ Life sciences    ☐ Behavioural & social sciences    ☒ Ecological, evolutionary & environmental sciences

For a reference copy of the document with all sections, see [nature.com/documents/nr-reporting-summary-flat.pdf](nature.com/documents/nr-reporting-summary-flat.pdf)

# Ecological, evolutionary & environmental sciences study design

All studies must disclose on these points even when the disclosure is negative.

| | |
|---|---|
| Study description | A metagenomics study on the discovery and characterization of novel microbial gene families with significant functional, ecological and evolutionary vcalue. |
| Research sample | A collection of over 140,000 MAGS collected from various public studies. |
| Sampling strategy | N/A |
| Data collection | 149,842 medium and high-quality metagenome assembled genomes (MAGs) and single-amplified genomes (SAGs), alongside 19,642 reference genomes from isolated and fully sequenced species. This set includes over 400 million gene predictions and was assembled |

by unifying five distinct data sources spanning 82 habitats: two MAG collection spanning thousands of samples from diverse origins (GEM and GMGC), a comprehensive human gut catalog (UHGG), a global ocean catalog (OMD),  and the GTDB r95 reference database (Table S1).

| Timing and spatial scale | N/A |
| Data exclusions | Low quality MAGs and spurious sequences removed. |
| Reproducibility | All available data and scripts to reproduce our analyses are provided as supplementary material and as an online repository. |
| Randomization | N/A |
| Blinding | N/A |

Did the study involve field work?  ☐ Yes  ☒ No

# Reporting for specific materials, systems and methods

We require information from authors about some types of materials, experimental systems and methods used in many studies. Here, indicate whether each material, system or method listed is relevant to your study. If you are not sure if a list item applies to your research, read the appropriate section before selecting a response.

## Materials & experimental systems

| n/a | Involved in the study |
|---|---|
| ☒ | ☐ Antibodies |
| ☒ | ☐ Eukaryotic cell lines |
| ☒ | ☐ Palaeontology and archaeology |
| ☒ | ☐ Animals and other organisms |
| ☒ | ☐ Clinical data |
| ☒ | ☐ Dual use research of concern |
| ☒ | ☐ Plants |

## Methods

| n/a | Involved in the study |
|---|---|
| ☒ | ☐ ChIP-seq |
| ☒ | ☐ Flow cytometry |
| ☒ | ☐ MRI-based neuroimaging |

## Plants

| Seed stocks | N/A |
| Novel plant genotypes | N/A |
| Authentication | N/A |

