## [Peer Review File · Nature]

Manuscript Title: Functional and evolutionary significance of unknown genes from uncultivated taxa

Reviewer Comments & Author Rebuttals

Reviewer Reports on the Initial Version:

Referees' comments:

Referee #1 (Remarks to the Author):

Rodriguez del Rio et al analyzed hundreds of thousands metagenomes, from diverse habitats to identify 413,335 protein families with unknown function. The authors present an elegant discovery pipeline for identifying 'orphan proteins' including a comprehensive annotation step with known databases and by incorporation of conservation, purifying selection and gene expression information. Further, the authors describe a method to assign putative functions to orphan genes and use it to annotate about one-fifth of them.

Significance:

Using a gene clustering-based approach, the authors provide a catalog of more than 400,000 protein families, a three-fold increase over the number of known bacterial orthologous groups to date. The authors predict the potential function of ~20% of them by genomic context analysis and provide an online resource to explore the genes and their context. The authors argue that these data and this method, upon their availability, could help to reduce a substantial limitation in the microbial functional genomics field. This limitation is that a high fraction of existing open reading frames in metagenomes have no annotated function and are 'hypothetical genes'. They 'de-orphan' the functions of some of these clustered gene families using genomic context analysis - and thus are able to provide some potential functions for unknown genes. This is certainly a step forward and will enhance existing databases.

Overall Impression:

Overall, this work appears to use appropriate tools and has been carried out in a rigorous manner. The work is focused on an important issue - which is that functional analyses of metagenomic data usually suffer from annotating the functions and thus comparing the functions of a small percentage (~15%) of all genes. Thus, studies that conclude that "these communities are taxonomically different but functionally the same" are ignoring the vast majority of genes within a community, and these are likely erroneous conclusions.

While this work focuses on an important topic - there are some areas for improvement. The writing would benefit from improvement in clarity and presentation - at times, the presentation of the findings seemed overly complicated. Also, many of the findings are presented as very novel - but these types of ideas (clustering open reading frames into gene families and using genomic neighborhood to ascertain function) have been previously implemented and considered in the literature. A more careful delineation of what is new and what has already been demonstrated

would be helpful to the reader. While an important step forward, the work presented does have some limitations: First, while their work makes predictions about the potential role of various genes, there are no experiments carried out to demonstrate the specific utility of these assignments, nor are there efforts to present orthogonal ways to test the validity of these functional predictions. I appreciate that these may be out of the scope of this purely computational manuscript. Second, while not a specific limitation, the clustering and genomic-neighborhood based approach described in this paper is solid but not entirely novel, having been used by other groups in previous work that is well cited in the manuscript. Third, it would have been much more powerful to see an example of how this new database could be used to empower a much improved functional analysis of a metagenomic dataset from the literature.

Major comments:

1. The author's workflow illustrated in Figure 1A provides a useful pipeline for both annotating proteins and for identifying orphan protein-coding gene families. However, I believe that their effort can potentially better serve the microbial functional genomics community by two further simple steps to make the data and workflow more digestible and accessible:
 - a. Making a user friendly tool to identify and predict function of orphan genes in a genome/MAG. This might be based on both assigning a novel protein to one of the protein clusters identified in this paper AND by genomic context based on that single genome/bin. The authors did provide the scripts for their analyzes here, but these are yet neither documented nor user-friendly.
 - b. Publish data with protein clusters, their sequences and a putative function for all the protein clusters (with confidence levels) which can be later independently used for homology-based annotation of proteins.
2. As mentioned, the authors elegantly assigned functions to protein clusters in order to filter-in orphan genes. However, the majority of protein families (the exact number is missing) do have annotated function. This dataset of probably millions of protein clusters with putative function can be valuable to the community, for further functional genomic studies.
3. Putative functional role was assigned by reconstruction of the genomic neighborhood for ~75K protein families. An estimation of the performance of this method would be helpful, specially because of the use of some arbitrary cutoffs ($e < 10^{-3}$, coverage > 50%, +/- 3 genes, conservation score > 0.9 etc) and can be done using known, functionally annotated proteins and their genomic context.
4. It would be nice to see a detailed analysis of the 5 novel protein families embedded in the Nitrogen cycle operons. How big are these proteins? What are their predicted folds/structures? What might they be doing? Are there any ways that you can add to the analysis in Figure 2b and 2c?
5. The authors highlight how an improved ability to annotate gene function can improve/change outcomes of metagenomic studies - why not actually carry out such an analysis? For example, in the original HMP studies, a conclusion was drawn that communities (between individuals) varied taxonomically, but were "functionally the same". Using the improved database that the authors develop could help challenge that (likely wrong) dogma. This type of analysis would make a big difference in convincing the reader of the utility and application of their findings.

Minor comments:

1. Synapomorphic proteins families - identifying synapomorphic protein clusters, with stronger purifying selection indeed sheds light on genes with innovative function for their lineage. The

authors suggest that ~0.2% (980/413K) of the orphan proteins are synapomorphic of phyla, classes or orders. This raises the question whether synapomorphism is equally common in functionally annotated and orphan protein families.

2. Clustering and identifying novel protein families suggest these proteins suggest these are not orphan proteins that actually have orthologs in different, probably uncultivated, organisms.

However, synapomorphic protein families represent newly emerged genes with probably lineage specific function. Can the authors suggest a hypothesis of how these coding genes evolved?

3. Please add the number of proteins/clusters to Figure 1A for each initial database and step.

4. It seems that ~15% of the multi-domain unknown proteins reside within viral contigs. This finding illustrates viral-dependent genetic flow, between domains and it would be very interesting if the authors could point on these viral contigs.

5. Additional information regarding the protein clusters would be helpful. What is the fraction of clusters with genes assembled from “pure” habitats (in example from host associated bacteria), what is the fraction that is mixed. Same for proteins with assigned function. I suspect we are biased for assigning functions to host associated bacteria and whether the genetic context method suffers from these biases as well.

6. Figure 1C is not clear. The authors do not refer to it in the main text and shortly describe it in the caption. Is it within cluster amino acid identity?

7. Fig 3C x axis title should read “Lineage specificity” (instead of lineage).

8. The authors term orphan protein clusters as orphan and uncultivated proteins interchangeably, I believe that orphan/unannotated.

9. Identifying conserved operons- the authors look for at least one contiguous functional annotated gene on the same strand. Did the authors set maximal distance (in bp) between the annotated and the orphan genes?

10. “hundreds of high-quality metagenome-assembled genomes are publicly available”- more than hundreds are available. This should be corrected.

11. A limitation of this method is its inability to describe novel functions, worth discussing this limitation with future perspective in the discussion.

Referee #2 (Remarks to the Author):

The manuscript “Functional and evolutionary significance of unknown genes from uncultivated taxa” introduces a catalogue and online resource for previously unknown genes from microorganisms that have not been cultured under laboratory conditions. I particularly enjoyed the part about synapomorphic genes. However, I have some major issues with the article that I am summarizing here.

Major issues:

- The title promises the functional significance of unknown genes from uncultivated taxa – however, I do not see any major functional predictions of unknown genes in the manuscript. This needs to be addressed as this would be the major advance here.

- The online resource (<http://novelfams.cgmlab.org>) is very problematic to use. Searches by taxon name only provide the result “No family found. We are unable to find any families matching the given criteria. Please try again!” – I entered even taxa names from the paper and got no result. The

advanced search is not working. Overall, I was unable to use the resource and it needs to be substantially improved to really make the presented data usable and thus citable.

- A clear distinction to previous work, e.g., by JGI regarding protein family clustering and potential function (several papers in previous years) or those examples mentioned below (<https://www.nature.com/articles/s41586-021-04233-4> and <https://www.nature.com/articles/s41467-019-12171-z>) is necessary in the manuscript. In particular, the global distribution of the protein families does not seem novel; similarly, the existence of distinct protein families in Patescibacteria have been published previously, too.

Minor issues:

Line 44: Please rephrase and state that environmental genomics – genome-resolved metagenomics and single cell sequencing – was responsible for the availability of genomes of uncultivated taxa. Currently single cell sequencing is not mentioned in this context.

Line 49: Those studies cannot predict if a gene is functional as they are based off gene prediction (unless you mean a different way of predicting that a gene is functional). I don't get the sense behind the fact that genes are functional but evolutionary untraceable. Maybe functional should be functionally and the adverb of untraceable in the sentence?

Line 55: please change meagenomic sequencing to environmental genomics

Line 75: “of thousands of uncultured organisms” – should be genomes of thousands of uncultured organisms since you are analyzing the genomes rather than the organisms. Moreover, these are population genomes rather than genomes of organisms, something that should be made clear somewhere in the introduction since they likely have never existed as such in nature unless you focus on SAGs.

Line 83: “fully sequenced species” are these genomes of isolates? If so, please say so.

Line 84: Will this set be made publicly available?

Line 103: VOGs also contain host genes that also occur in viruses. Wouldn't that remove potential genes frequently present in hosts?

Line 107: Please define previously unknown [...] protein families. There have been studies exploring protein families in general and I'm wondering if your previously unknown protein families have not yet been detected by others before. For example, some of the authors also conducted this study <https://www.nature.com/articles/s41586-021-04233-4>, which includes clustering of prokaryotic genes into gene families. A discrimination of the “previously unknown” protein defined here and those presented in the earlier article is necessary.

Line 148: I think the finding that Patescibacteria (aka CPR) contain differing protein families than regular bacteria has been previously published: <https://www.nature.com/articles/s41467-019-12171-z> - so what makes the current study novel in that regard?

Line 157-176: It appears that most of the information was published previously in <https://www.nature.com/articles/s41586-021-04233-4> - so I'm uncertain what's new here.

Line 177-19X: I'm missing a link to Figure 4A and Figure 4B. Somehow these two panels are not mentioned in the text at all. They are however necessary to understand this paragraph.

Line 374: I do not understand the rationale behind using a bray crutis index here. One of the main ideas of bray Curtis index is to avoid creating similarities based on the absence of datapoints – however, the absence of certain (synapomorphic) protein families is a major discriminator according to the data presented.

Figure 4A: Please explain “average genomes” and “average taxa” in the figure legend.

Referee #3 (Remarks to the Author):

In this manuscript, the authors identify new gene families from a large collection of metagenome-assembled genomes (MAGs) and single-cell-assembled genomes (SAGs) from across multiple environments. They find that many of these gene families are clade-specific (as expected), and that many are restricted to particular environments.

The technical methods are for the most part well-chosen (but see below), and the technical achievement is impressive. The main limitations of this manuscript as currently presented are in interpretation and further use of the data.

The following points should be addressed:

1. An important limitation is that the gene families were searched only against GTDB, which contains bacteria and archaea only, not eukaryotes. Are any of the novel gene families found in eukaryotes but missed by searching only against a database that does not contain them? It is possible that this is addressed in the database search section for inclusion criteria, but if so it needs to be clarified what taxa each of these databases contain.
2. How confident are the authors in the quality of the assemblies used as input? Do the results hold if only a smaller but higher quality subset of SAGs and MAGs is used?
3. The motivation for doing the work is that a large fraction of genes in metagenomic samples remains uncharacterized. I was therefore surprised not to see some measure of what fraction of genes in metagenomic samples from different habitats is recruited to the new gene catalog that the authors have created.
4. The claim that "the role of unknown protein families
134 may not be limited to accessory molecular functions but could also involve central metabolic
132 processes" does not follow from "Of those, 1,344 families share a genomic
132 context with known and highly conserved marker genes related to energy production or
xenobiotic 133 compound degradation pathways" because xenobiotic degradation pathways are
typically considered to be accessory functions.
5. "Habitat" is not well defined. The authors should use some combination of clustering of the metagenomic data and an ontology or controlled vocabulary to define the habitats. "Ecological breadth" is very hard to assess when some habitats are likely to be much more similar than others.
6. The analysis of synapomorphic proteins, although interesting, is difficult to interpret without comparison of the results of the same analysis on known protein families. This should be provided.
7. Given the availability of structure prediction methods, it is surprising that no structure prediction was attempted.

8. No sense is given of how much this paper expands what we know about the microbial protein universe, what fraction of the way to completing this task we are given this work, whether particular environments need more focused sampling effort than others, etc. Adding this would be very useful for helping readers assess the level of advance.

Minor:

"Linage" in Figg. 3C should be "Lineage"

Author Rebuttals to Initial Comments:

REBUTTAL LETTER

[REDACTED] :

[REDACTED]

[REDACTED]

[REDACTED]

Referees' comments:

Referee #1 (Remarks to the Author):

Rodriguez del Rio et al analyzed hundreds of thousands metagenomes, from diverse habitats to identify 413,335 protein families with unknown function. The authors present an elegant discovery pipeline for identifying 'orphan proteins' including a comprehensive annotation step with known databases and by incorporation of conservation, purifying selection and gene expression information. Further, the authors describe a method to assign putative functions to orphan genes and use it to annotate about one-fifth of them.

Significance:

Using a gene clustering-based approach, the authors provide a catalog of more than 400,000 protein families, a three-fold increase over the number of known bacterial orthologous groups to date. The authors predict the potential function of ~20% of them by genomic context analysis and provide an online resource to explore the genes and their context. The authors argue that these data and this method, upon their availability, could help to reduce a substantial limitation in the microbial functional genomics field. This limitation is that a high fraction of existing open reading frames in metagenomes have no annotated function and are 'hypothetical genes'. They 'de-orphan' the functions of some of these clustered gene families using genomic context analysis - and thus are able to provide some potential functions for unknown genes. This is certainly a step forward and will enhance existing databases.

We thank the reviewer for this encouraging summary of our work.

Overall Impression:

Overall, this work appears to use appropriate tools and has been carried out in a rigorous manner. The work is focused on an important issue - which is that functional analyses of metagenomic data usually suffer from annotating the functions and thus comparing the functions of a small percentage (~15%) of all genes. Thus, studies that conclude that "these communities are taxonomically different but functionally the same" are ignoring the vast majority of genes within a community, and these are likely erroneous conclusions.

Thank you for your suggestion. We have now added a new section to the manuscript where we specifically explore the impact of including our novel gene families in functional profiling, multidimensional analysis, predictor training, and biomarker identification. More details can be found in the following sections.

While this work focuses on an important topic - there are some areas for improvement. The writing would benefit from improvement in clarity and presentation - at times, the presentation of the findings seemed overly complicated. Also, many of the findings are presented as very novel - but these types of ideas (clustering open reading frames into gene families and using genomic neighborhood to ascertain function) have been previously implemented and considered in the literature. A more careful delineation of what is new and what has already been demonstrated would be helpful to the reader.

We have now thoroughly revised the manuscript to improve the clarity of our findings. Both the Introduction and Discussion sections have been largely rewritten to better cover and contextualize previous work and to more clearly delineate the novelties of our study.

While an important step forward, the work presented does have some limitations:

First, while their work makes predictions about the potential role of various genes, there are no experiments carried out to demonstrate the specific utility of these assignments, nor are there efforts to present orthogonal ways to test the validity of these functional predictions. I appreciate that these may be out of the scope of this purely computational manuscript.

We thank the reviewer for this comment. Indeed, experimental follow-ups require a lot of additional time and effort and was originally considered outside the scope of this computational work. Note that, by definition, none of the novel gene families have been detected in cultured organisms, which poses a major challenge for the validation of their function.

However, we have now included two experimental validations guided by our functional predictions, which provide compelling examples of the validity of our functional assignments, support the quality of the catalog, and illustrate its impact on different areas of research. We have also included additional benchmarks, cross-validations, and confidence values for our functional predictions. Below is a list of the specific actions taken:

1. We have experimentally validated two of our functional predictions, demonstrating the value of the resource for discovering new genes with important functions:
 - a. *"NOV3845Y, found in at least two species from the unknown genus UBA8309, was predicted to be involved in the bacterial chemotaxis pathway with a confidence score of 0.8. Given its position in a gene cluster canonically responsible for chemotaxis, we hypothesized that NOV3845Y may play a role in cell motility in response to environmental changes. To test our hypothesis, we performed swimming chemotaxis assays on a Escherichia coli (E. coli) strain heterologously expressing NOV3845Y. To account for plasmid burden, we used another E. coli strain transformed with the empty vector (pBSK) as a control. Results showed that the strain expressing NOV3845Y displayed a larger swimming halo than the pBSK control strain (Fig. S2), confirming the role of NOV3845Y in chemotaxis signaling and cell motility."*

- b. "*NOVOQR9B*, a gene family of short peptides, found in 39 *Faecalibacillus* genomes, is systematically conserved in a genomic region predicted to be associated with antibiotic resistance (3 conserved resistance neighbors within the CARD database). In addition, *NOVOQR9B* contains antimicrobial signal motifs (antimicrobial probability by Macrel(Santos-Júnior et al. 2020): 0.545). We synthesized the protein and tested, in vitro, its potential antimicrobial activity against *Paenibacillus polymyxa*, *Bacillus subtilis*, *Lactobacillus* sp., *E. coli*, and *Pseudomonas putida* (three Gram-positive and two Gram-negative bacterial species). *NOVOQR9B* was capable of effectively inhibiting growth of all three Gram-positive bacteria, but it had no effect on the Gram-negative species (Fig. S3). "

2. We now show that our genomic context analysis is robust for establishing functional associations of novel gene families:
 - a. We provide a new comprehensive benchmark of our genomic context method based on the analysis of already known genes in our metagenomics catalog. This allowed us to calibrate what is the minimum evidence of synteny and genomic context conservation required to predict functional associations for each KEGG pathway from the MAGs and SAGs in our dataset, and to provide confidence values for our predictions.
 - b. We compared the functional predictions of Miller *et al.*'s method (Miller, Stern, and Burstein 2022) and our genomic context conservation approach, finding that they agreed on their functional associations for 81% of the high confidence predictions.
 - c. We also included functional predictions based on the genomic analysis published by Miller *et al.* during the review process of our article.

3. In addition, we now provide functional predictions based on structural similarity computed de novo with CollabFold and FoldSeek. This provides an additional independent source of more specific functional predictions that complements the genomic context analysis. Interestingly, 38.8% of the novel families with high confidence KEGG pathway predictions based on genomic context were also supported by structural alignments.

Second, while not a specific limitation, the clustering and genomic-neighborhood based approach described in this paper is solid but not entirely novel, having been used by other groups in previous work that is well cited in the manuscript.

Indeed, clustering and synteny analysis are computational methods that have been extensively used and validated in the past, including previous work to quantify novelty (Nayfach et al. 2021; Sberro et al. 2019; Almeida et al. 2021), unify known and unknown environmental genes (Salazar et al. 2019; Coelho et al. 2022; Vanni et al. 2022), or the identification of lineage-specific novel protein families defining new lineages (Méheust et al. 2019). We have now delineated more clearly what is the status of these methodologies, and highlighted the novelty of our work.

For example, we recognise that previous work has applied similar clustering strategies to group metagenomic sequences into gene families and used genomic context reconstruction for functional prediction of unknown genes, albeit at different scales and/or for different purposes. This is now discussed in the introduction, where we emphasize that the novelty of our work is not in the application of the clustering algorithm itself, but in the curation of the resulting clusters to obtain a catalog of novel gene families with high evolutionary, functional, and ecological significance. This is, to our knowledge, a work that has never been addressed at the scale and depth presented in our paper.

Note that our goal was not to quantify functionally unknown environmental genes in bulk, nor to compare the known versus unknown sequence space, like performed in other recent analyses (Vanni et al. 2021). Instead, we were primarily motivated by the general concerns that have so far prevented the integration of unknown sequences from the microbial dark matter into

standard metagenomic surveys, like the presence of potential sequencing artifacts, contaminations, gene fragments, false orphans, pseudogenes, or just sequences of insufficient quality or evidence. In fact, our catalog represents only 5% of all the gene family clusters inferred by the original clustering analysis and it is willfully biased towards most promising candidates in terms of biological significance at different levels, as we hope to demonstrate in our paper.

Still, given the size and extent of our original dataset, we show that this 5% of novelty represents a significant increase in the number of prokaryotic orthologous groups described to date, which (as we also demonstrate in the revised manuscript) has a clear impact on functional profiling and other related analyses. These ideas are now more clearly reflected in the paper.

Finally, we have also clarified in the text that the use of genomic context to predict gene function is a well-known approach in comparative genomics, and that it has been previously applied to the functional prediction of environmental sequences. The most current example would be the recent paper on the use of natural language processing models as method for functional prediction, by Miller and colleagues (Miller, Stern, and Burstein 2022), which was published during our review process. In the revised version of our manuscript, we extended the functional annotations based on their approach and used their dataset to provide orthogonal validation of our predictions. Our results show that high-confidence predictions are 81% in agreement between the two methods. In addition, 22,898 (5.6%) of our novel gene families received additional functional annotations with their approach, which we have also incorporated as additional annotations in our resource.

Finally, all our functional predictions are now provided with confidence scores and can be visualized using our online database browser, which implements unique tools for interactively exploring the conservation of genomic context across MAGs and SAGs and aims at facilitating future hypothesis-driven analysis.

Third, it would have been much more powerful to see an example of how this new database could be used to empower a much improved functional analysis of a metagenomic dataset from the literature.

This is indeed an excellent suggestion. Initially, we considered such a practical application to be beyond the scope of this work, but we have now developed a framework that allows for the easy exploitation of our catalog in future metagenomic surveys. In particular, we have incorporated the sequence signatures of all novel OGs into the eggNOG-mapper software database, a widely used software for functional annotation and profiling of genomes and metagenomes. Thus, in addition to functional profiling of metagenomic samples based on eggNOG and KEGG ortholog groups, it is now possible to identify the presence of our novel gene families in custom genomes or metagenomes and access their functional predictions (which are reported in the eggNOG-mapper output).

In the revised version of our manuscript, we now demonstrate and discuss the predictive potential of novel gene family abundance profiles for discriminating metagenomic samples by habitat and condition. We found that profiling analyses based solely on the relative abundances of novel gene families are capable of discriminating samples from different habitats,

mammalian guts, human body sites, and even human guts from different populations (Fig. S9 and S10).

In addition, the impact of incorporating novel gene families into functional analysis pipelines is now illustrated by means of the re-analysis of 575 human gut samples from a clinical study (Wirbel et al. 2019) involving 285 colorectal cancer (CRC) patients and 290 controls from 5 different populations. We employed our enhanced version of eggNOG-mapper to identify novel gene families in all these samples and estimated their relative abundances. Then, using the same stringent thresholds as in Wirbel *et al.*, we found 69 over-abundant novel gene families in CRC patients, some of which were functionally predicted to be associated with cell adhesion and/or biofilm. We also show that predictors trained solely on the relative abundances of novel gene families across samples not only provided similar accuracy to state-of-the-art functional models, but also slight improvements (1-2%) over current profiling methods when discriminating disease vs control conditions.

Major comments:

1. The author's workflow illustrated in Figure 1A provides a useful pipeline for both annotating proteins and for identifying orphan protein-coding gene families. However, I believe that their effort can potentially better serve the microbial functional genomics community by two further simple steps to make the data and workflow more digestible and accessible:

a. Making a user friendly tool to identify and predict function of orphan genes in a genome/MAG. This might be based on both assigning a novel protein to one of the protein clusters identified in this paper AND by genomic context based on that single genome/bin. The authors did provide the scripts for their analyzes here, but these are yet neither documented nor user-friendly.

We agree that this would be extremely useful. As explained above, we have now incorporated the novel protein families into the EggNOG-mapper software and in its online version at <http://eggnog-mapper.embl.de/FESNOV>. In addition, the predicted functional associations (e.g. genomic context, structural alignments and intrinsic features) of each detected novel family are reported in the eggNOG-mapper output, so the tool can be used for functional prediction and profiling.

We have also improved the documentation of the novel gene family discovery and characterization pipeline (<https://github.com/AlvaroRodriguezDelRio/nov-fams-pipeline>) to allow researchers to run it independently on any collection of genomes. The pipeline is obviously complex and requires large computational resources, but the fundamental steps and scripts are now documented and should be readily usable by any bioinformatician. In fact, during the review process of this paper, we collaborated with two other projects that used our pipeline to discover new gene families and functions out of newly described lineages, which illustrates the usefulness of the methods developed:

<https://www.biorxiv.org/content/10.1101/2023.07.03.547478v1.abstract>
<https://www.nature.com/articles/s41467-022-34388-1>

b. Publish data with protein clusters, their sequences and a putative function for all the protein clusters (with confidence levels) which can be later independently used for homology-based annotation of proteins.

The sequences of all protein clusters and their associated data, including functional predictions with eggNOG-mapper, are now available in the download section of the online resource (<https://novelfams.cgmlab.org/>). Functional predictions for the novel gene families have also been consolidated into a single table, including prediction sources and confidence scores.

In addition, we now provide many original intermediate files to ensure the reproducibility of our analyses and to facilitate access to other types of data that may be useful for future studies. This includes, genome assemblies, functional predictions, protein structures, etc.

2. As mentioned, the authors elegantly assigned functions to protein clusters in order to filter-in orphan genes. However, the majority of protein families (the exact number is missing) do have annotated function. This dataset of probably millions of protein clusters with putative function can be valuable to the community, for further functional genomic studies.

We now provide all sequences and clusters in the raw files available in the download section of <https://novelfams.cgmlab.org/>. Functional annotations inferred with eggNOG-mapper for the entire set (400M proteins) along with the cluster membership of each sequence are also available for download. The total number of sequences and clusters obtained at each step of our novel gene family discovery pipeline is now clarified in the text and in Figure 1A.

3. Putative functional role was assigned by reconstruction of the genomic neighborhood for ~75K protein families. An estimation of the performance of this method would be helpful, specially because of the use of some arbitrary cutoffs ($e < 10^{-3}$, coverage > 50%, +/- 3 genes, conservation score > 0.9 etc) and can be done using known, functionally annotated proteins and their genomic context.

Thanks for the suggestion. As mentioned in our previous comments, we have now computed a comprehensive benchmark for both estimating the accuracy of functional prediction on a per-pathway basis, and assigning a confidence score to each of our predictions.

To do this, we tested the ability of our method to predict KEGG pathways of already known genes out of environmental genomes, allowing us to calibrate the minimum synteny and functional genomic context conservation required to achieve accurate predictions for each KEGG pathway. As expected, the level of conservation required to achieve high confidence predictions was different for each functional category. Pathway-specific parameters were then applied to assign a confidence score to each functional prediction.

From all novel gene families, 203,617 received low confidence (confidence score < 0.5), 52,793 medium confidence (confidence \geq 0.5), and 4,349 high confidence (confidence \geq 0.9) predictions.

We also extended our predictions with those predicted by Miller *et al.* (2023) by mapping our novel gene families against their dataset (leading to the annotation of 22,898 gene families). In addition, we predicted protein structures and alignments against the PDB and Uniprot

databases, allowing us to annotate 56,609 gene families based on structural similarities. Interestingly, 38.8% of the novel families with both high-confidence genomic context predictions and significant structural alignments were assigned to the same pathway by both methods.

Overall, 32.4% of the FESNov catalog received at least one functional prediction from each of the methods and sources used. All annotations are now provided in a consolidated table and have been integrated into eggNOG-mapper for functional analyses of custom genomes and metagenomes.

4. It would be nice to see a detailed analysis of the 5 novel protein families embedded in the Nitrogen cycle operons. How big are these proteins? What are their predicted folds/structures? What might they be doing? Are there any ways that you can add to the analysis in Figure 2b and 2c?

We carried out extensive work to characterize these novel families, including cloning them into different bacterial strains. Unfortunately, the cloning experiments resulted in very low expression or solubility, or in lethal phenotypes, which prevented us from conducting further studies. However, the fact that the function of these novel families could not be validated in vitro is not surprising, given that they are derived from uncultivated species with very different genetic backgrounds. Therefore, although they may well be related to nitrogen metabolism, we could not confirm it experimentally.

Nonetheless, although distant homology by PSI-BLAST analysis did not yield any hits to known genes from the RefSeq genomes, the new addition of structural-based analysis in our data, allowed us to gain additional evidence about these sequences. We particularly found that NOV5WD8W, one of the novel gene families reported in Figure 2B, is adjacent to the NosZ-encoding gene for nitrous oxide reductase and has a significant structural similarity (PLDDT=82, FoldSeek score=263) to the periplasmic copper metallochaperone cusF.

NosZ is a periplasmic multicopper enzyme that reduces N₂O to N₂ in the final step of bacterial denitrification. The maturation of Apo-NosZ to functional NosZ in the periplasmic space requires a complex machinery of accessory transporters, reductases and chaperones located around the cytoplasmic membrane and in the periplasmic space. Interestingly, a key component of this machinery is NosL, a copper-metal chaperone thought to be involved in copper supply to NosZ (www.pnas.org/cgi/doi/10.1073/pnas.1903819116), which we could not find in the same operon. The significant folding similarity of NOV5WD8W to a copper metallochaperone suggests that this new family may play a role in the transfer of copper to NosZ.

We have now added a paragraph in the main text explaining the details of this interesting example, which illustrates the usefulness of our predictions for hypothesis-driven analyses. We have also added a length scale to all schematic representations of the FESNov gene families.

5. The authors highlight how an improved ability to annotate gene function can improve/change outcomes of metagenomic studies - why not actually carry out such an analysis? For example, in the original HMP studies, a conclusion was drawn that communities (between individuals) varied taxonomically, but were "functionally the same". Using the improved database that the

authors develop could help challenge that (likely wrong) dogma. This type of analysis would make a big difference in convincing the reader of the utility and application of their findings.

We appreciate the suggestion, and we agree that demonstrating the impact of using our catalog for profiling metagenomic samples would indeed be a powerful way of proving its utility. We have addressed this issue from two different angles:

First, we now demonstrate that it is possible to discriminate between environments and even individuals using solely the relative abundance matrices of novel gene families across thousands of samples (Fig. S9, 10). We believe this is remarkable, as the signal used for such analysis comes entirely from unknown genes uniquely found in uncultivated organisms. The results were supported by both multidimensional analysis and SVM predictors (e.g. precision over 90% for various human populations), and reinforces the idea that the curated set of novel gene families in our catalog carry a great ecological value.

On the other hand, as mentioned by the reviewer, a lack of resolution in functional profiling based on known genes has been reported in the past. However, this is typically associated with the use of broad functional categories such as COG categories or KEGG pathways or modules. Finer entities, like KEGG Orthologs (KO) or eggNOG groups, which would be equivalent to our novel gene families, have been shown to provide better resolution in discriminating between environments and populations. For instance, state-of-the-art functional profiling tools such as HUMAnN2 (<https://www.nature.com/articles/s41592-018-0176-y>) shows that changes in the abundances of KEGG KOs were associated with sample temperature in oceans. Similarly, we believe that the lack of resolution in HMP functional profiling analyses is mostly due to the use

of general functional terms. Based on our analyses, we can only report the same (good) accuracy of novel gene families (putative novel orthologous groups) compared to fine grained KEGG Orthologs terms (KOs). For instance, both functional profiling based on KEGG KOs and novel gene families provide good resolution to discriminate habitats and conditions using multidimensional analyses methods such as tSNE.

Second, we believe that the use of novel gene families would have a significant and greater impact on discovering potential biomarkers that are specific to certain environments or conditions. To verify this hypothesis, we analyzed 575 human gut samples from a clinical study (Wirbel et al. 2019) that included 285 patients with colorectal cancer and 290 controls from five distinct populations. Our improved version of eggNOG-mapper was used to identify new gene families in all the samples, for which we estimated their relative abundances.

We then utilized logistic regression and random forest methods to train predictors based on the relative abundances of new gene families across all samples. The novel gene families in CRC samples exhibited remarkable discrimination power (AUC = 0.74), and improved the predictor's accuracy by 1% compared to the use of known genes alone. Furthermore, applying the same stringent thresholds as in Wirbel *et al.*, we looked for significantly over-abundant novel gene families in patients diagnosed with CRC. Notably, we discovered 69 unique gene families with a pronounced enrichment in CRC samples, which could serve as promising biomarkers for CRC diagnosis. Using our taxonomic and functional annotations, we noticed that a few of these new biomarker families have links to organisms already connected to CRC, and they show functional association with cell adhesion and biofilm formation. Together, this example illustrates how the simple inclusion of our catalog in standard metagenomic pipelines might have a direct impact on the results, providing novel targets and fresh mechanistic insights.

Minor comments:

1. Synapomorphic proteins families - identifying synapomorphic protein clusters, with stronger purifying selection indeed sheds light on genes with innovative function for their lineage. The authors suggest that ~0.2% (980/413K) of the orphan proteins are synapomorphic of phyla, classes or orders. This raises the question whether synapomorphism is equally common in functionally annotated and orphan protein families.

This is an interesting question. We have now broadened the analysis to include all clusters, using several known examples to validate our results:

"A gene family is considered synapomorphic if it is nearly ubiquitous within a certain clade (high coverage), while being mostly absent from any other species (high specificity). Under such criteria, very few clusters should be expected at basal taxonomic levels. In fact, using a cutoff of 70% coverage and 90% specificity, we only detected 34,370 synapomorphic clusters at high-level taxonomic ranks (phylum, class or order; Table S9), which included functionally characterized families well known for being characteristic of certain lineages. For instance, the photosystem P840 reaction center protein (pscD), and the chlorosome envelope protein C (csmC) were detected as synapomorphic at the chlorobia bacterial class (Hauska et al. 2001; Li and Bryant 2009). Similarly, the photosystems I subunits psaD and psaF, the photosystem II oxygen-evolving enhancer protein psbO and the NHD1 complex subunit (ndhN) were also detected as synapomorphic for the entire cyanobacteria phylum (Table S9).

In order to identify the most prominent synapomorphies in uncultivated lineages, we scanned the entire set of 12,132,456 clusters originally inferred with a minimum of 90% clade coverage and 100% clade specificity. Although these strict thresholds offer very little tolerance for the potential incompleteness of environmental genomes used, and/or old horizontal gene transfers, we still found 2,210 clusters that could be considered synapomorphic at the phylum, class, or order level (Table S10). Interestingly, almost half of these high-level synapomorphic clusters (1,034 families) were included in our final FESNov catalog (Table S10), supporting the idea that our set is greatly enriched in evolutionarily relevant families. "

Moreover, we verified that our scoring and detection methods strongly correlate with other lineage specificity measurements (e.g. AnnoTree (Mendler et al. 2019)), which we have used as an orthogonal validation and it is now mentioned in the methods section of the manuscript.

With the new analysis, we realized that the main set of synapomorphies shown in the paper represent the most prominent cases, but, indeed, other highly relevant families could be found with slightly lower scores. Therefore we have now included in the manuscript a full table of novel and known gene families with their corresponding 'synapomorphic' scores across all taxonomic levels.

2. Clustering and identifying novel protein families suggest these proteins suggest these are not orphan proteins that actually have orthologs in different, probably uncultivated, organisms. However, synapomorphic protein families represent newly emerged genes with probably lineage specific function. Can the authors suggest a hypothesis of how these coding genes evolved?

That's a very important question that would be very difficult to answer with our current data. We believe that the hypotheses for the origin of this specific set of novel gene families (synapomorphic) are not different from those proposed for other genes in prokaryotes and eukaryotes (Kaessmann 2010). Those include duplication/amplification models (Romero and Palacios 1997), rapid evolution and sequence divergence following transfer events (Ochman, Lawrence, and Groisman 2000), gene fusions (Yanai, Wolf, and Koonin 2002), emergence of new functions from pseudogenes or non-coding regions (Zheng and Gerstein 2007), and more.

In fact, we do not believe that any of those particular models is the main contributor to our catalog of novel genes in general, or to the synapomorphic set in particular. Different

explanations may actually apply to different instances in our set. For example, in the synapomorphic set we found an obvious case of a highly divergent homolog of the ccmD gene (heme transported protein D) in the HRBIN17 class (with 26 genomes and 7 species). Although the sequences of this family (NOV0DFBV) did not produce significant blast hits against RefSeq, the genomic context strongly supports the idea of being a distant homolog of ccmD. However, we also found hundreds of cases for which no sequence or structural similarities could be found (even when we relaxed our homology thresholds). These new sequences and their respective novel protein folds could have originated from any of the above models. We have included a few sentences in the paper noting the fact that many novel families might represent new protein folds and functions, but we think proposing potential evolutionary origins would be highly speculative with our current data and should be analyzed on a per-family basis.

3. Please add the number of proteins/clusters to Figure 1A for each initial database and step.

We computed a total of 12,132,456 gene families, among which 7,052,473 are novel (no hit to PfamA, PfamB, Refseq and Egnog). We have now incorporated this information into Figure 1A and into the main text.

4. It seems that ~15% of the multi-domain unknown proteins reside within viral contigs. This finding illustrates viral-dependent genetic flow, between domains and it would be very interesting if the authors could point on these viral contigs.

We now provide all contigs and their annotation as viral or plasmid as estimated by the Plasflow and Seeker software, respectively.

5. Additional information regarding the protein clusters would be helpful. What is the fraction of clusters with genes assembled from “pure” habitats (in example from host associated bacteria), what is the fraction that is mixed.

20,434 novel gene families are specific (pure) to only one of the 157 habitats. After habitat grouping into 10 habitat groups, 28,074 novel gene families are habitat specific. We have included this information to the manuscript, and provided extended data as supplementary tables, presenting which families are specific of which habitats.

Same for proteins with assigned function. I suspect we are biased for assigning functions to host associated bacteria and whether the genetic context method suffers from these biases as well.

This is an interesting check that we have now performed. We found that there is indeed a small habitat bias in functional annotation performance. Setting the confidence value of our genomic context method to 90%, we were able to annotate 1.2% of the novel families detected in host-associated samples but only 0.9% of pure free-environmental samples. We have included this information in the methods section.

6. Figure 1C is not clear. The authors do not refer to it in the main text and shortly describe it in the caption. Is it within cluster amino acid identity?

This figure is now referenced in the main text and the legend was clarified. Indeed, the figure represents within cluster amino acid identity.

7. Fig 3C x axis title should read "Lineage specificity" (instead of lineage).

Thanks for noticing, this has now been changed.

8. The authors term orphan protein clusters as orphan and uncultivated proteins interchangeably, I believe that orphan/unannotated.

The manuscript has been revised to eliminate the term "orphan protein cluster", while also clarifying that our gene families are not singletons nor orphans. We clarify that they could erroneously be described as such if compared solely to the genomic databases of isolated/cultured organisms, but not when compared to other uncultivated taxa. In fact, our analysis indicates that many of these genes are not orphan when analyzed in the context of other uncultivated taxa.

9. Identifying conserved operons- the authors look for at least one contiguous functional annotated gene on the same strand. Did the authors set maximal distance (in bp) between the annotated and the orphan genes?

In the original manuscript, we only considered novel families located less than 100 nucleotides away from the known functional gene as it is now indicated in the methods section. For this revision's new benchmark, we also added a maximum nucleotide distance parameter to calculate prediction accuracy per pathway.

10. "hundreds of high-quality metagenome-assembled genomes are publicly available"- more than hundreds are available. This should be corrected.

Thanks. This was an error, we meant "hundreds of thousands".

11. A limitation of this method is its inability to describe novel functions, worth discussing this limitation with future perspective in the discussion.

We now discuss the limitations of the functional annotation methods to provide very detailed insights about potential novel functions. However, the general-level functional associations inferred are proven useful for guiding hypothesis-driven experimental assays, as we have demonstrated with the validation of two of our predictions. In addition, we now discuss the potential of structural based inferences to provide more detailed predictions, and highlight the fact that most of the new families identified seem to encode for new folds.

Referee #2 (Remarks to the Author):

The manuscript "Functional and evolutionary significance of unknown genes from uncultivated taxa" introduces a catalogue and online resource for previously unknown genes from microorganisms that have not been cultured under laboratory conditions. I particularly enjoyed the part about synapomorphic genes. However, I have some major issues with the article that I am summarizing here.

Thanks for the comments and suggestions. In the following, we explain how we addressed all the major issues reported.

Major issues:

- The title promises the **functional significance of unknown genes from uncultivated taxa – however, I do not see any major functional predictions of unknown genes in the manuscript**. This needs to be addressed as this would be the major advance here.

In our work, we meant to identify functionally significant unknown gene families (e.g. related to important biological processes), but not necessarily new major molecular functions. This is a subtle difference that we tried to reflect in the title, but we could of course rephrase if needed.

Note that, the unknown genes and families in our catalog are probably a mix of cases involving new major molecular functions, analogs and distant undetectable homologs. For instance, we cannot discard the possibility that unknown genes with undetectable sequence or structural similarity to anything known, still perform similar molecular functions than other known genes. However, we are confident that all these new sequences are highly relevant from a functional and evolutionary perspective, given their conservation level, strong purifying selection, expression evidence and synteny conservation on functionally important genomic regions. As part of this revision, we also computed structural predictions and alignments for all novel gene families, highlighting the fact that a large proportion encodes for putative new folds.

Nonetheless, we appreciate this reviewer's comment and we agree that providing examples of new molecular activities derived from the catalog would significantly improve our work. In the revised version of the manuscript, we have included two experimental validations and one structural based analysis in the manuscript, which illustrate the potential of hypothesis-driven characterization of new functions based on our predictions. We performed the following validations (response also provided to to review #1):

1. *"NOV3845Y, found in at least two species from the unknown genus UBA8309, was predicted to be involved in the bacterial chemotaxis pathway with a confidence score of 0.8. Given its position in a gene cluster canonically responsible for chemotaxis, we hypothesized that NOV3845Y may play a role in cell motility in response to environmental changes. To test our hypothesis, we performed swimming chemotaxis assays on a Escherichia coli (E. coli) strain heterologously expressing NOV3845Y. To account for plasmid burden, we used another E. coli strain transformed with the empty vector (pBSK) as a control. Results showed that the strain expressing NOV3845Y displayed a larger swimming halo than the pBSK control strain (Fig. S2), confirming the role of NOV3845Y in chemotaxis signaling and cell motility."*

2. "NOVOQR9B, a gene family of short peptides, found in 39 *Faecalibacillus* genomes, is systematically conserved in a genomic region predicted to be associated with antibiotic resistance (3 conserved resistance neighbors within the CARD database). In addition, NOVOQR9B contains antimicrobial signal motifs (antimicrobial probability by Macrel(Santos-Júnior et al. 2020): 0.545). We synthesized the protein and tested, in vitro, its potential antimicrobial activity against *Paenibacillus polymyxa*, *Bacillus subtilis*, *Lactobacillus* sp., *E. coli*, and *Pseudomonas putida* (three Gram-positive and two Gram-negative bacterial species). NOVOQR9B was capable of effectively inhibiting growth of all three Gram-positive bacteria, but it had no effect on the Gram-negative species (Fig. S3). "

3. The gene family NOV5WD8W, one of the novel gene families reported in Figure 2B, was predicted to be associated with Nitrogen metabolism. We carried out extensive work to characterize it experimentally, including cloning into different bacterial strains. Unfortunately, the cloning experiments resulted in very low expression or solubility, or in lethal phenotypes, which prevented us from conducting further studies. The fact that the function of these novel families could not be validated in vitro is not surprising, given that they are derived from uncultivated species with very different genetic backgrounds. However, although distant homology by PSI-BLAST analysis did not yield any hits to known genes from the RefSeq genomes, the new addition of structural-based analysis allowed us to gain additional evidence. We particularly found that NOV5WD8W, one of the novel gene families reported in Figure 2B, is adjacent to the *NosZ*-encoding gene for nitrous oxide reductase and has a significant structural similarity (PLDDT=82, FoldSeek score=263) to the periplasmic copper metallochaperone *cusF*. *NosZ* is a periplasmic multicopper enzyme that reduces N₂O to N₂ in the final step of bacterial denitrification. The maturation of Apo-*NosZ* to functional *NosZ* in the periplasmic space requires a complex machinery of accessory transporters, reductases and chaperones located around the cytoplasmic membrane and in the periplasmic space. Interestingly, a key component of this machinery is *NosL*, a copper-metal chaperone thought to be involved in copper supply to *NosZ* (www.pnas.org/cgi/doi/10.1073/pnas.1903819116), which we could not find in the same operon. The significant folding similarity of NOV5WD8W to a copper metallochaperone (Fig S4) suggests that this new family may play a role in the transfer of copper to *NosZ*.

Besides describing new molecular functions in different contexts, these experiments and case examples illustrate the great potential of our resource and functional predictions to guide the characterization of new important functions in the unknown microbial space.

- The online resource (<http://novelfams.cgmlab.org>) is **very problematic to use**. Searches by taxon name only provide the result “No family found. We are unable to find any families matching the given criteria. Please try again!” – I entered even taxa names from the paper and got no result. The advanced search is not working. Overall, I was unable to use the resource and it needs to be substantially improved to really make the presented data usable and thus citable.

Thank you for reporting these issues. We have now completely redesigned the online resource to provide more intuitive and integrated navigation of the data. Originally, the web was primarily designed as an online repository with basic search and browse capabilities. The new online database has been improved so it serves as an interactive browser of the catalog:

- The search panel is now tolerant of partial term searches, case-insensitive, and much faster. We suspect that taxon name searches were not working for the reviewer because the system was expecting exact GTDB names (e.g. including the __s or __p tags). This is now fixed and all searches allow partial matches and use an autocomplete context menu to make navigation easier.
- The search system is now more powerful, yet easier to use. For example, the same search panel can be used to find specific novel gene families by their code (NOVxxxxx), by their predicted functional associations, or by their taxonomic content.
- Search results are now presented in a more integrative way, with all available data for each novel gene family displayed as cards with further links to expand the information.
- The online database now provides a dedicated menu for easy access to predefined lists of novel gene families that correspond to the different analyses mentioned in the manuscript. For example, these predefined lists include the subsets of novel gene families with high confidence functional predictions, embedded in biosynthetic gene clusters, containing antimicrobial peptide signals, considered synapomorphic, and more.
- We have also added a sequence-based search that allows fast, blast-like searches against the catalog. This feature is based on fast and sensitive searches against a database of 400 million HMM models, which are also available for download.
- We now provide a download section with all raw, intermediate, and final data used and produced by this work. This includes raw assemblies, sequences, clustering results, functional mappings, and many other files that have been standardized and organized to ensure reproducibility and further analysis.

- A clear distinction to previous work, e.g., by JGI regarding protein family clustering and potential function (several papers in previous years) or those examples mentioned below (<https://www.nature.com/articles/s41586-021-04233-4> and <https://www.nature.com/articles/s41467-019-12171-z>) is necessary in the manuscript. In particular, the global distribution of the protein families does not seem novel; similarly, the existence of distinct protein families in Patescibacteria have been published previously, too.

Thank you for the comment. This issue was also mentioned by reviewer 1 and addressed in full in this revision. (For convenience, parts of our answer to reviewer 1 are also included here)

Indeed, clustering and synteny analysis are computational methods that have been extensively used and validated in the past, including previous work to quantify novelty (Nayfach et al. 2021; Sberro et al. 2019; Almeida et al. 2021), unify known and unknown environmental genes (Salazar et al. 2019; Coelho et al. 2022; Vanni et al. 2021), or the identification of lineage-specific novel protein families defining new lineages (Méheust et al. 2019). We have now delineated more clearly what is the status of this methodology, and highlighted the novelty of our work.

For example, we recognise that previous work has applied similar clustering strategies to group metagenomic sequences into gene families and used genomic context reconstruction for functional prediction of unknown genes, albeit at different scales and/or for different purposes. This is now discussed in the introduction, where we emphasize that the novelty of our work is not in the application of the clustering algorithm itself, but in the curation of the resulting clusters to obtain a catalog of novel gene families with high evolutionary, functional, and ecological significance. This is, to our knowledge, a work that has never been addressed at the scale and depth presented in our paper.

In addition, we explicitly acknowledge that novel protein clusters have been studied in the past for particular lineages, such as Patecibacteria. Hence, we highlight more clearly that the novelty of our work is not only extending the phylogenetic and ecological scope of such analyses, but carefully delineating which of those novel gene families are more relevant from an evolutionary (i.e. synapomorphies), ecological (e.g. prevalent, habitat specific) and functional (associated with important pathways) standpoint.

Regarding previous work by us and others, we now reflect in the manuscript that, although the global distribution of microbial protein families (known and unknown) has been indeed proven to be vast, it is mostly ecologically narrow at the global scale. In Coelho *et al.*, we showed that the majority of genes in the global microbiome group into a small number of families (Coelho et al. 2022) and usually represent neutral or nearly neutral variants (Coelho et al. 2022) (i.e. no positive selection pressures). Similarly, it has been reported that functionally unknown genes tend to group mostly into species-specific clusters and are ecologically narrow (Vanni et al. 2022).

By meticulously curating the vast space of unknown functions, we compiled a set of families where the majority (59.4%) can be detected in more than ten samples from at least two different habitats (Table S12), a pattern that is maintained even after consolidating habitat annotations into a less detailed ecological categorization of just ten biomes. In other words, the gene families in our catalog— which were not identified based on ecological criteria— invert the general ecological pattern observed for the bulk set of environmental genes and protein families.

These clarifications are now included in the main manuscript, which highlights the novelty of our findings and the context of previous work.

Minor issues:

Line 44: Please rephrase and state that environmental genomics – genome-resolved metagenomics and single cell sequencing – was responsible for the availability of genomes of uncultivated taxa. Currently single cell sequencing is not mentioned in this context.

Thanks for the comment. We now refer to “environmental DNA sequencing” and "environmental genomes" in the manuscript.

Line 49: Those studies cannot predict if a gene is functional as they are based off gene prediction (unless you mean a different way of predicting that a gene is functional). I don't get the sense behind the fact that genes are functional but evolutionary untraceable. Maybe functional should be functionally and the adverb of untraceable in the sentence?

We understand the confusion, this was a mistake in the word *functional*, which should have been written as "*functionally*". In the new manuscript, we formulate the same idea in a (hopefully) more clearly way, referring to these unknown sequences as genes without homologs in cultured organisms:

"The fraction of environmental genes lacking homologs in cultured organisms, hereafter referred to as "unknown", varies from approximately 25%(Almeida et al. 2021) to nearly 50%(Nayfach et al. 2021), depending on the methodology and specific environment. This translates to millions of novel genes whose functional aspects and ecological and evolutionary significance remain unknown."

Line 55: please change meagenomic sequencing to environmental genomics

We have now changed this in the text.

Line 75: “of thousands of uncultured organisms” – should be genomes of thousands of uncultured organisms since you are analyzing the genomes rather than the organisms. Moreover, these are population genomes rather than genomes of organisms, something that should be made clear somewhere in the introduction since they likely have never existed as such in nature unless you focus on SAGs.

We agree that the vast majority of our data comes from MAGs, rather than SAGs, and therefore we should better refer to the (population) genomes of uncultured organisms, rather than to individual organisms. We have now changed this sentence accordingly and reviewed the rest of the text to avoid inconsistencies.

" (...) and have only become feasible with the recent publication of hundreds of thousands of environmental genomes."

Line 83: “fully sequenced species” are these genomes of isolates? If so, please say so.

Thank you, we now refer to reference genomes from isolated and fully sequenced species.

" (...) alongside 19,642 reference genomes from isolated and fully sequenced species."

Line 84: Will this set be made publicly available?

Yes, the entire dataset, including intermediate files derived from our various analyses are now available in the download section of the online database.

Line 103: VOGs also contain host genes that also occur in viruses. Wouldn't that remove potential genes frequently present in hosts?

Yes, indeed. But we aimed at selecting prokaryotic genes not present in reference databases, so host gene families potentially matching the pVOGs database were actually removed in earlier stages of the pipeline. Moreover, pVOGs include a very very low frequency of host genes (mean of 0.011) and therefore most of the entries could be considered viral-only sequences (mean viral quotient = 0.88).

Line 107: Please define previously unknown [...] protein families. There have been studies exploring protein families in general and I'm wondering if your previously unknown protein families have not yet been detected by others before. For example, some of the authors also conducted this study <https://www.nature.com/articles/s41586-021-04233-4>, which includes clustering of prokaryotic genes into gene families. A discrimination of the "previously unknown" protein defined here and those presented in the earlier article is necessary.

We agree that these words were not precise enough. We have now changed the sentence to refer to "previously uncharacterized" novel gene families, as all raw data used in our study were actually downloaded from public repositories and therefore could be present in other datasets and studies. For instance, the MAGs from our previous work (GMGC catalog) were directly included in our analysis, and our estimations are that around 237k of the curated novel families in our catalog could be detected in the GMGC set. However, none of these sequences were identified as novel, not characterized in any specific way.

Line 148: I think the finding that Patescibacteria (aka CPR) contain differing protein families than regular bacteria has been previously published: <https://www.nature.com/articles/s41467-019-12171-z> - so what makes the current study novel in that regard?

We have now better contextualized our results regarding the Patescibacteria phylum. (Méheust et al. 2019) highlights that Patescibacteria has a distinctive gene repertoire, and that CPR enriched gene families have mostly poor functional annotations. In contrast, we highlight the high proportion of novel gene families in our filtered catalog that co-occur in the Patescibacteria genome (and not only in the overall lineage). We also point out that these families are enriched in transmembrane sequences that distinguish the clade, confirming previous findings (Méheust et al. 2019) as we now acknowledge in the text.

Line 157-176: It appears that most of the information was published previously in <https://www.nature.com/articles/s41586-021-04233-4> - so I'm uncertain what's new here.

We understand the confusion. Actually, the only overlap between Coelho *et al.* and this study is the clustering step, which was run using the same software and with similar parameters.

In Coelho *et al.*, we never distinguished between known and unknown protein clusters, so none of the conclusions (including ecological distributions and specificity) refer to unknown protein families from uncultivated taxa. We also used a much smaller set of samples and metagenomes.

In this study, we combined data from Coelho *et al.* with other large datasets. The ecological distribution was estimated *de novo* by mapping novel gene families against a different and larger collection of samples and habitats as those used in Coelho *et al.* Note that the ecological patterns inferred in our study came from analyzing over 64,000 public samples (in contrast to the ~13,000 used in Coelho *et al.*)

Most importantly, all our analyses refer to a relatively small set of unknown families that are unique from uncultivated taxa and meet other criteria, which actually reverts the patterns described in our previous work for the full set of gene clusters. For instance, the novel families included in our catalog invert the narrow ecological patterns reported for overall gene families (Coelho *et al.* 2022), and unknown gene clusters in particular (Vanni *et al.* 2021). We consider this a remarkable result, as none of our selection filters included ecological parameters. These important distinctions and additional discussion is now included in different parts of the manuscript.

Line 177-19X: I'm missing a link to Figure 4A and Figure 4B. Somehow these two panels are not mentioned in the text at all. They are however necessary to understand this paragraph.

We have included references to all sub-panels of figure 4 in the text.

Line 374: I do not understand the rationale behind using a bray curtis index here. One of the main ideas of bray Curtis index is to avoid creating similarities based on the absence of datapoints – however, the absence of certain (synapomorphic) protein families is a major discriminator according to the data presented.

We used bray curtis indexes for measuring the similarity of the samples where a gene family was detected. As this was not very relevant for the work, and to avoid misunderstandings, we have now eliminated this part.

Figure 4A: Please explain “average genomes” and “average taxa” in the figure legend.

This is now explained in the figure legend.

Referee #3 (Remarks to the Author):

In this manuscript, the authors identify new gene families from a large collection of metagenome-assembled genomes (MAGs) and single-cell-assembled genomes (SAGs) from across multiple environments. They find that many of these gene families are clade-specific (as expected), and that many are restricted to particular environments.

The technical methods are for the most part well-chosen (but see below), and the technical achievement is impressive. The main limitations of this manuscript as currently presented are in interpretation and further use of the data.

We thank the reviewer for their encouraging comments about our work.

The following points should be addressed:

1. An important limitation is that the gene families were searched only against GTDB, which contains bacteria and archaea only, not eukaryotes. Are any of the novel gene families found in eukaryotes but missed by searching only against a database that does not contain them? It is possible that this is addressed in the database search section for inclusion criteria, but if so it needs to be clarified what taxa each of these databases contain.

Thanks for the suggestion. All the environmental genomes included in our study were of medium and high quality and assigned to prokaryotic organisms. However, we agree that there is always a risk of contamination and quimerization. We were aware of these issues and the eukaryotic control suggested by the reviewer was indeed performed in our original pipeline (but perhaps not made sufficiently clear in the text and methods).

One of the exclusion criteria from our catalog was having a significant match to either eggNOG, PFAM, or RefSeq, which are broad databases containing eukaryotic sequences. We are therefore confident that there are no obvious eukaryotic genes in our catalog. In addition, all of our gene families must be present in at least three genomes from two different species, which reduces the likelihood of artifacts.

However, a valid concern would be that these three databases may not cover the full eukaryotic biodiversity. To address this possibility, we have now compared our catalog with EukProt, a database that collects eukaryotic proteomes across the tree of life, covering all major lineages. We found only 1,127 novel gene families (0.27%) in our catalog with significant hits to EukProt, most of which are very distant homologs (33% identity on average), with only 12 families showed more than 90% similarity.

Although these gene families could be the result of genomic chimeras or assembly artifacts, they could also: i) represent distant prokaryotic homologs of interest for the study of eukaryogenesis (e.g., we found that several of these families were present in Asgard genomes); ii) involve cross-kingdom horizontal gene transfer events, or iii) represent prokaryotic contaminations (or actual transfers) in the eukaryotic genomes of EukProt.

Therefore, we decided to keep these 1,127 cases in the catalog, but flag them for careful consideration (Table S16).

2. How confident are the authors in the quality of the assemblies used as input? Do the results hold if only a smaller but higher quality subset of SAGs and MAGs is used?

We were very strict with both the quality filters of the source data and the supporting evidence required to include gene families in our catalog.

First, all MAGs included in our study were originally tagged as medium or high quality. Nevertheless, we recalculated the completeness and contamination values of all MAGs and even recalculated the open reading frames of MAGs to account for their genetic code differences (for example, we noticed that genes from the Gracilibacteria or Mycoplasma lineages were incorrectly predicted under the standard codon table in some public datasets).

Overall, 98.9% of the MAGs and SAGs used in our study have a contamination level of less than 5%. The median contamination was actually 0.77%, and the median completeness was ~93%.

In addition, to estimate the potential effect of removing medium-quality genomes, we now calculated how many families are supported by high-quality genomes. Overall, 90.33% of the novel families in our catalog have at least 1 sequence member from a high-quality genome.

Moreover, note that all gene families in our catalog must be supported by at least three genomes from two different species, which greatly reduces the possibility of having gene families based on chimeric or erroneous gene calls. Other filters based on the detection of negative selection and coding sequences also increased the quality of the final set of gene families analyzed here.

3. The motivation for doing the work is that a large fraction of genes in metagenomic samples remains uncharacterized. I was therefore surprised not to see some measure of what fraction of genes in metagenomic samples from different habitats is recruited to the new gene catalog that the authors have created.

This is a very valid statement. The novel gene families in our catalog represent 4,366,641 genes, ~1% from the original dataset (400M). When using our catalog to profile metagenomic samples from different sources, we observed an increase between 1 and 3 percent in the annotation of sequences.

However, we believe that the value of our catalog is not that much about how many genes can be recruited in future studies, but about their potential biological relevance and impact on downstream analyses. For instance, all the families in our catalog are under strong purifying selection, found in more than one species and contain new putative protein domains. Also, all of them have been characterized from a functional (when possible), evolutionary, structural and ecological perspective, which allowed us to analyze them beyond quantitative approaches. These aspects are now included in the manuscript with the new sections that demonstrate the impact of incorporating our novel gene families into the standard metagenomic workflows.

4. The claim that "the role of unknown protein families may not be limited to accessory molecular functions but could also involve central metabolic processes" does not follow from "Of those, 1,344 families share a genomic context with known and highly conserved marker genes related to energy production or xenobiotic 133 compound degradation pathways" because xenobiotic degradation pathways are typically considered to be accessory functions.

Thanks. We have now modified the text and do not refer to degradation pathways as central metabolism.

5. "Habitat" is not well defined. The authors should use some combination of clustering of the metagenomic data and an ontology or controlled vocabulary to define the habitats. "Ecological breadth" is very hard to assess when some habitats are likely to be much more similar than others.

Indeed, we are aware of this problem, which is common to many multi-habitat studies. In fact, although some of the original datasets of MAGs used in our study provided habitat information (e.g., GMGC and GEM), we did not rely on such classification for our ecological analyses for the same reasons outlined by the reviewer.

Instead, we mapped all of our novel gene families to a comprehensive set of ~63k public metagenomic samples for which we had previously curated and standardized the habitat information. The habitat classification of these samples, and thus the one used in our ecological analyses, consists of a hierarchical ontology of habitat names, from more general to more specific. This allowed us to calculate ecological breadth for habitats at the same ontology level, partially mitigating the problem raised by the reviewers. The 157 habitat categories used are shown in Supplementary Table 10. In addition, a paper describing this habitat ontology in more detail is under review in the upcoming NAR database issue (<https://spire.embl.de/>). We will be happy to provide access to the paper for this review process if needed.

Furthermore, we understand that some habitats in this controlled vocabulary may still be considered more general than others. Therefore, to ensure that our main distributional results are consistent, we repeated our analyses using a more general ecological classification of the sample sources, consisting of ten high-level classifications of environments that could be considered biomes. The same trends and patterns were found and are also reported in the manuscript.

6. The analysis of synapomorphic proteins, although interesting, is difficult to interpret without comparison of the results of the same analysis on known protein families. This should be provided.

We have now broadened the analysis to include all clusters, using several known examples to validate our results:

"A gene family is considered synapomorphic if it is nearly ubiquitous within a certain clade (high coverage), while being mostly absent from any other species (high specificity). Under such criteria, very few clusters should be expected at basal taxonomic levels. In fact, using a cutoff of 70% coverage and 90% specificity, we only detected 34,370 synapomorphic clusters at high-level taxonomic ranks (phylum,

class or order; Table S9), which included functionally characterized families well known for being characteristic of certain lineages. For instance, the photosystem P840 reaction center protein (pscD), and the chlorosome envelope protein C (csmC) were detected as synapomorphic at the chlorobia bacterial class (Hauska et al. 2001; Li and Bryant 2009). Similarly, the photosystems I subunits psaD and psaF, the photosystem II oxygen-evolving enhancer protein psbO and the NHD1 complex subunit (ndhN) were also detected as synapomorphic for the entire cyanobacteria phylum (Table S9).

In order to identify the most prominent synapomorphies in uncultivated lineages, we scanned the entire set of 12,132,456 clusters originally inferred with a minimum of 90% clade coverage and 100% clade specificity. Although these strict thresholds offer very little tolerance for the potential incompleteness of environmental genomes used, and/or old horizontal gene transfers, we still found 2,210 clusters that could be considered synapomorphic at the phylum, class, or order level (Table S10). Interestingly, almost half of these high-level synapomorphic clusters (1,034 families) were included in our final FESNov catalog (Table S10), supporting the idea that our set is greatly enriched in evolutionarily relevant families. "

Moreover, we verified that our scoring and detection methods strongly correlate with other lineage specificity measurements (e.g. AnnoTree (Mendler et al. 2019)), which we have used as an orthogonal validation and it is now mentioned in the methods section of the manuscript.

With the new analysis, we realized that the main set of synapomorphies shown in the paper represent the most prominent cases, but, indeed, other highly relevant families could be found with slightly lower scores. Therefore we have now included in the manuscript a full table of novel and known gene families with their corresponding 'synapomorphic' scores across all taxonomic levels.

7. Given the availability of structure prediction methods, it is surprising that no structure prediction was attempted.

We agree that structural predictions would be an excellent addition to our resource and functional analyses.

During the review process of this paper, a huge catalog of structures predicted using natural language models was published (Lin et al. 2023). Therefore, our first attempt was to map our catalog to their database and retrieve their already predicted structures so that we could use them for functional analysis based on structural alignments. Unfortunately, only 33.6% of our novel gene families were found in their database.

Therefore, we addressed *de novo* computations of all our families using the alphaFold approach. In total, we computed 389,638 structures (96% of our catalog) using ColabFold, which took several months of computation on our GPU clusters. The remaining 4% of families were attempted but failed due to technical issues in the software. Of these structures, 226,991 (56.17%) received a high confidence score (PLDDT \geq 70) and were used to extend the functional predictions of the novel gene families by computing structural alignments against PDB and Uniprot using FoldSeek. In total, we found 56,609 significant structural alignments,

which we used to functionally annotate the corresponding novel gene families. Interestingly, high-confidence genomic context-based predictions and structural predictions showed 38.8% agreement at the KEGG pathway level, providing additional support for the inferred functional associations.

On the other hand, the lack of high quality structural predictions and structural alignments for more than 86% of the families suggests that novel gene families may encode as yet unknown protein folds.

The new data and results are now reported and discussed in the manuscript.

8. No sense is given of how much this paper expands what we know about the microbial protein universe, what fraction of the way to completing this task we are given this work, whether particular environments need more focused sampling effort than others, etc. Adding this would be very useful for helping readers assess the level of advance.

In our paper, we highlighted the fact that our catalog doubles the number of prokaryotic orthologous groups described to date. We now discuss the fact that, despite representing a relatively small dataset over the entire unknown sequence space, our catalog was extremely strict in its filters, leaving out millions of unknown sequences that could become more reliable or informative in the future. We now explicitly discuss this in the paper:

"Our work focused on identifying a highly curated set of so far uncharacterised gene families, for which we systematically addressed common problems like sequencing artifacts, contaminations, gene fragments, false orphans, pseudogenes, or sequences of insufficient quality or evidence—typical issues that have hampered the comparative analysis of the so-called microbial dark matter(Quince et al. 2017; Escudeiro, Henry, and Dias 2022). This meant focussing the analysis on a relatively small portion of the unknown sequences from the original dataset. However, despite its relatively small size, the FESNov catalog represents a threefold increase in the total number of known prokaryotic orthologous groups described to date, thus significantly expanding our ability to profile and compare metagenomic samples."

Minor:

"Linage" in Figg. 3C should be "Lineage"

- This typo is now fixed.

References

- Almeida, Alexandre, Stephen Nayfach, Miguel Boland, Francesco Strozzi, Martin Beracochea, Zhou Jason Shi, Katherine S. Pollard, et al. 2021. "A Unified Catalog of 204,938 Reference Genomes from the Human Gut Microbiome." *Nature Biotechnology* 39 (1): 105–14.
- Coelho, Luis Pedro, Renato Alves, Álvaro Rodríguez Del Río, Pernille Neve Myers, Carlos P. Cantalapiedra, Joaquín Giner-Lamia, Thomas Sebastian Schmidt, et al. 2022. "Towards the Biogeography of Prokaryotic Genes." *Nature* 601 (7892): 252–56.
- Escudeiro, Pedro, Christopher S. Henry, and Ricardo P. M. Dias. 2022. "Functional Characterization of Prokaryotic Dark Matter: The Road so Far and What Lies Ahead."

- Current Research in Microbial Sciences* 3 (August): 100159.
- Hauska, G., T. Schoedl, H. Remigy, and G. Tsiotis. 2001. "The Reaction Center of Green Sulfur bacteria(1)." *Biochimica et Biophysica Acta* 1507 (1-3): 260–77.
- Kaessmann, Henrik. 2010. "Origins, Evolution, and Phenotypic Impact of New Genes." *Genome Research* 20 (10): 1313–26.
- Li, Hui, and Donald A. Bryant. 2009. "Envelope Proteins of the CsmB/CsmF and CsmC/CsmD Motif Families Influence the Size, Shape, and Composition of Chlorosomes in *Chlorobaculum Tepidum*." *Journal of Bacteriology* 191 (22): 7109–20.
- Lin, Zeming, Halil Akin, Roshan Rao, Brian Hie, Zhongkai Zhu, Wenting Lu, Nikita Smetanin, et al. 2023. "Evolutionary-Scale Prediction of Atomic-Level Protein Structure with a Language Model." *Science* 379 (6637): 1123–30.
- Méheust, Raphaël, David Burstein, Cindy J. Castelle, and Jillian F. Banfield. 2019. "The Distinction of CPR Bacteria from Other Bacteria Based on Protein Family Content." *Nature Communications*. <https://doi.org/10.1038/s41467-019-12171-z>.
- Mendler, Kerrin, Han Chen, Donovan H. Parks, Briallen Lobb, Laura A. Hug, and Andrew C. Doxey. 2019. "AnnoTree: Visualization and Exploration of a Functionally Annotated Microbial Tree of Life." *Nucleic Acids Research* 47 (9): 4442–48.
- Miller, Danielle, Adi Stern, and David Burstein. 2022. "Deciphering Microbial Gene Function Using Natural Language Processing." *Nature Communications* 13 (1): 5731.
- Nayfach, Stephen, Simon Roux, Rekha Seshadri, Daniel Udway, Neha Varghese, Frederik Schulz, Dongying Wu, et al. 2021. "A Genomic Catalog of Earth's Microbiomes." *Nature Biotechnology* 39 (4): 499–509.
- Ochman, H., J. G. Lawrence, and E. A. Groisman. 2000. "Lateral Gene Transfer and the Nature of Bacterial Innovation." *Nature* 405 (6784): 299–304.
- Quince, Christopher, Alan W. Walker, Jared T. Simpson, Nicholas J. Loman, and Nicola Segata. 2017. "Shotgun Metagenomics, from Sampling to Analysis." *Nature Biotechnology* 35 (9): 833–44.
- Romero, D., and R. Palacios. 1997. "Gene Amplification and Genomic Plasticity in Prokaryotes." *Annual Review of Genetics* 31: 91–111.
- Salazar, Guillem, Lucas Paoli, Adriana Alberti, Jaime Huerta-Cepas, Hans-Joachim Ruscheweyh, Miguelangel Cuenca, Christopher M. Field, et al. 2019. "Gene Expression Changes and Community Turnover Differentially Shape the Global Ocean Metatranscriptome." *Cell* 179 (5): 1068–83.e21.
- Santos-Júnior, Célio Dias, Shaojun Pan, Xing-Ming Zhao, and Luis Pedro Coelho. 2020. "Macrel: Antimicrobial Peptide Screening in Genomes and Metagenomes." *PeerJ* 8 (December): e10555.
- Sberro, Hila, Brayon J. Fremin, Soumaya Zlitni, Fredrik Edfors, Nicholas Greenfield, Michael P. Snyder, Georgios A. Pavlopoulos, Nikos C. Kyrpides, and Ami S. Bhatt. 2019. "Large-Scale Analyses of Human Microbiomes Reveal Thousands of Small, Novel Genes." *Cell* 178 (5): 1245–59.e14.
- Vanni, Chiara, Matthew S. Schechter, Silvia G. Acinas, Albert Barberán, Pier Luigi Buttigieg, Emilio O. Casamayor, Tom O. Delmont, et al. 2021. "Unifying the Known and Unknown Microbial Coding Sequence Space." *bioRxiv*. <https://doi.org/10.1101/2020.06.30.180448>.
- . 2022. "Unifying the Known and Unknown Microbial Coding Sequence Space." *eLife* 11 (March). <https://doi.org/10.7554/eLife.67667>.
- Wirbel, Jakob, Paul Theodor Pyl, Ece Kartal, Konrad Zych, Alireza Kashani, Alessio Milanese, Jonas S. Fleck, et al. 2019. "Meta-Analysis of Fecal Metagenomes Reveals Global Microbial Signatures That Are Specific for Colorectal Cancer." *Nature Medicine* 25 (4): 679–89.
- Yanai, Itai, Yuri I. Wolf, and Eugene V. Koonin. 2002. "Evolution of Gene Fusions: Horizontal Transfer versus Independent Events." *Genome Biology* 3 (5): research0024.
- Zheng, Deyou, and Mark B. Gerstein. 2007. "The Ambiguous Boundary between Genes and Pseudogenes: The Dead Rise Up, or Do They?" *Trends in Genetics: TIG* 23 (5): 219–24.

Reviewer Reports on the First Revision:

Referees' comments:

Referee #1 (Remarks to the Author):

Overall, this manuscript is much improved and I appreciate the authors' careful attention to the comments offered by this and other reviewers. I have only a few additional comments:

Major:

1. The manuscript could benefit from some careful editing for clarity and flow – there are several places in the manuscript where the wording is awkward or unclear. I have listed a few here – but there are others. As with my previous comments, I think this manuscript could benefit greatly from attention to simplifying the language used and being more clear/direct.
2. I sincerely appreciate the great lengths that the research team took to try and validate some of their computational predictions. This is to be applauded. Quite frankly, I am amazed that the motility phenotype was elicited with heterologous expression of a single gene in *E. coli*. I appreciate the plasmid control – an even better control would have been a mutagenized gene/scrambled gene or something along those lines. Given the focus on NOV3845Y (the motility related protein), I'd be curious to know some basic facts about this gene and recommend that this is included in the main manuscript: how long is the protein? What is its domain structure (helices, beta sheets), what does the predicted structure look like? What structural homologs does it have using FoldSeek? Also, what is the higher order known phylogeny of UBA8309? Gram positive? Gram negative? I see that in the figure (2c) it is noted to be in Proteobacteria – I think it is worth mentioning this in the written results also, since this will affect the reader's expectation regarding whether the testing of this gene in *E. coli* makes sense (which it does, given that the gene is in a Proteobacterium). Also, the fact that it is right in the middle of a cheBYAR operon makes it very highly likely to be a chemotaxis gene – and makes me very surprised that it hasn't been described in the past (and brings me back to the question of what structural homologs it has using FoldSeek).
3. With NOVOAR9B – would also like to know length of gene product, predicted secondary structure (I'd guess an amphipathic helix). This should be relatively easy to add. Also recommend adding the higher level phylogenetic information (Firmicutes) To the written results. It might be worth noting that the predicted AMP is next to an ABC transporter, and opining on this a bit. But not strictly necessary.
4. The online/interactive GUI is improved – but I wasn't able to get the search by gene function to work: <https://novelfams.cgmlab.org/>; also, I don't know what "Predicted as Antimicrobial Signal Peptide" means in the drop down menu of predefined sets.
5. The discussion would benefit from a dedicated paragraph focused on the limitations of this work.

Minor:

1. Line 22 – 'untraceable' – unusual word choice
2. Line 23 – While I appreciated that another reviewer prefers the term "environmental genome" to "metagenome" – I don't fully agree. Environmental genome (with genome used in the singular), implies to me a single genome, and I suspect that other readers will assume the same. At the end of the day, this is a decision for the authors and the editor, but I prefer 'environmental metagenomes',

for example. Again with Line 42 – I’m not sure that those who sequence host-associated microbiomes would consider their sequencing to be of “environmental DNA”.

3. Again, a matter of personal style – but overall, there is an excessive use of adverbs and adjectives – consider toning this down a bit and letting the reader decide what is ‘meticulously’ curated and ‘precisely’ distinguished.

4. Line 35 – I’m not sure what you mean by “greatly impact the standard metagenomics workflow” – recommend clarifying.

5. Line 37 – awkward wording “are capable of distinguishing between clinical conditions such as human gut samples from colorectal cancer”

6. Line 43 – what exactly is meant by “metabarcoding”?

7. Line 48 – I don’t really know what is meant by “a broad consensus prevails regarding the potential ...”. This sentence could be simplified and clarified.

8. Line 54 – Not sure “untapped” is quite accurate. For example, recent preprints have ‘tapped’ these sequences for the discovery of novel AMPs. Others have mined these sequences for novel Cas and gene editing/writing systems. I’m sure there are other interesting examples, as well. This should be noted and cited. Though I agree that the genetic repertoire of microbes on earth are ‘undertapped’.

9. Line 71 – how can a known gene family that is a species-specific cluster be an orphan?

10. In the results, please indicate how genes were identified in the collection of MAGs/SAGs and reference genomes. Was a standard annotation pipeline used for all, or were pre-provided annotations used. It is helpful for the reader to know this without having to look through the methods.

11. It might have been interesting to test some of the antibiotic resistance related gene families that were noted – perhaps this should be suggested in the discussion (I think it is out of the scope of the current manuscript).

12. Line 218 – don’t think the word ‘basis’ is necessary.

13. Line 327 – word choice: ‘candidates’

Referee #2 (Remarks to the Author):

The authors have fully addressed all my previous queries. I particularly enjoy the new search tool on their website and the clarity of words in sections I wasn’t able to comprehend before.

Congratulations to this great piece of work.

Author Rebuttals to First Revision:

Referees' comments:

Referee #1 (Remarks to the Author):

Overall, this manuscript is much improved and I appreciate the authors' careful attention to the comments offered by this and other reviewers. I have only a few additional comments:

Major:

1. The manuscript could benefit from some careful editing for clarity and flow – there are several places in the manuscript where the wording is awkward or unclear. I have listed a few here – but there are others. As with my previous comments, I think this manuscript could benefit greatly from attention to simplifying the language used and being more clear/direct.

Thanks, we have reviewed the text with the help of an editorial assistance service to simplify some sentences and use more direct language. For consistency, we have also replaced the term 'protein family' by 'gene family' all over the manuscript, as both terms were used indistinctly.

2. I sincerely appreciate the great lengths that the research team took to try and validate some of their computational predictions. This is to be applauded. Quite frankly, I am amazed that the motility phenotype was elicited with heterologous expression of a single gene in *E. coli*. I appreciate the plasmid control – an even better control would have been a mutagenized gene/scrambled gene or something along those lines. Given the focus on NOV3845Y (the motility related protein), I'd be curious to know some basic facts about this gene and recommend that this is included in the main manuscript: how long is the protein? What is its domain structure (helices, beta sheets), what does the predicted structure look like? What structural homologs does it have using FoldSeek? Also, what is the higher order known phylogeny of UBA8309? Gram positive? Gram negative? I see that in the figure (2c) it is noted to be in Proteobacteria – I think it is worth mentioning this in the written results also, since this will affect the reader's expectation regarding whether the testing of this gene in *E. coli* makes sense (which it does, given that the gene is in a Proteobacterium). Also, the fact that it is right in the middle of a cheBYAR operon makes it very highly likely to be a chemotaxis gene – and makes me very surprised that it hasn't been described in the past (and brings me back to the question of what structural homologs it has using FoldSeek).

Indeed, this seems to be an interesting new gene family involved in chemotaxis, which we are currently investigating in depth. As suggested, we have included some basic information about this gene family in the main manuscript, including explicit mention that the gene is embedded in the che operon. No reliable protein structures were obtained with CollabFold, nor significant hits against known proteins, which is also mentioned now in the main manuscript.

3. With NOVOAR9B – would also like to know length of gene product, predicted secondary structure (I'd guess an amphipathic helix). This should be relatively easy to add. Also recommend adding the higher level phylogenetic information (Firmicutes) To the written results. It might be work noting that the predicted AMP is next to an ABC transporter, and opining on this a bit. But not strictly necessary.

We have now added such information to the text, but decided not to speculate on its proximity to an ABC transporter, as manuscript length was already an issue.

4. The online/interactive GUI is improved – but I wasn't able to get the search by gene function to work: <https://novelfams.cgmlab.org/>; also, I don't know what "Predicted as Antimicrobial Signal Peptide" means in the drop down menu of predefined sets.

Search by gene function should be working now. Regarding "Predicted as Antimicrobial Signal Peptide", we meant gene families predicted as antimicrobial peptides. We have changed the term accordingly.

5. The discussion would benefit from a dedicated paragraph focused on the limitations of this work.

We have re-formatted the discussion paragraphs to reflect more clearly the limitations of our work:

- *The lack of specificity in functional predictions based on genomic context.*
- *The low number of protein structures with known foldings and structural similarity with known proteins*
- *The potential taxonomic re-classification of synapomorphies based on new discovered species and available genomes*
- *The low number of unknown sequences covered by the FESNov catalog (i.e. passing quality filters) relative to the total amount of unknown sequences in the original dataset.*

Minor:

1. Line 22 – ‘untraceable’ – unusual word choice

We have simplified the sentence and removed this word from the text.

2. Line 23 – While I appreciated that another reviewer prefers the term “environmental genome” to “metagenome” – I don’t fully agree. Environmental genome (with genome used in the singular), implies to me a single genome, and I suspect that other readers will assume the same. At the end of the day, this is a decision for the authors and the editor, but I prefer ‘environmental metagenomes’, for example. Again with Line 42 – I’m not sure that those who sequence host-associated microbiomes would consider their sequencing to be of “environmental DNA”.

We actually agree that "environmental DNA sequencing" and "environmental genome" (singular) could be misleading terms, so we have removed them from the text.

However, we kept the term "environmental genomes" (plural) when referring to both Metagenome Assembled Genomes (MAGs) and Single Amplified Genomes (SAGs), which is in our opinion a correct terminology.

3. Again, a matter of personal style – but overall, there is an excessive use of adverbs and adjectives – consider toning this down a bit and letting the reader decide what is ‘meticulously’ curated and ‘precisely’ distinguished.

Thanks for the suggestion. We have now simplified many sentences in the main text.

4. Line 35 – I’m not sure what you mean by “greatly impact the standard metagenomics workflow” – recommend clarifying.

We referred to the standard metagenomic profiling workflows and biomarker discovery pipelines. As this was specifically mentioned and clarified in the next sentence, we have removed the one mentioned by this reviewer.

5. Line 37 – awkward wording “are capable of distinguishing between clinical conditions such as human gut samples from colorectal cancer”

We have replaced the sentence to be more clear.

6. Line 43 – what exactly is meant by “metabarcoding”?

We refer to amplicon based sequencing for species identification. We think the term metabarcoding is quite accepted in the literature, so we decided to keep it.

7. Line 48 – I don’t really know what is meant by “a broad consensus prevails regarding the potential ...”. This sentence could be simplified and clarified.

We have simplified the sentence.

8. Line 54 – Not sure “untapped” is quite accurate. For example, recent preprints have ‘tapped’ these sequences for the discovery of novel AMPs. Others have mined these sequences for novel Cas and gene editing/writing systems. I’m sure there are other interesting examples, as well. This should be noted and cited. Though I agree that the genetic repertoire of microbes on earth are ‘undertapped’.

We have rephrased this paragraph to be more accurate.

9. Line 71 – how can a known gene family that is a species-specific cluster be an orphan?

The term "orphan" refers to genes that are species-specific. So, a cluster (gene family) containing several genes from several genomes, all of them classified as the same species (i.e. strains), could be considered an orphan gene family. Such clusters are very commonly obtained when processing metagenomics data. However, we explicitly excluded those cases from our analysis, focusing only on gene families detected in at least two different species (not orphan genes).

10. In the results, please indicate how genes were identified in the collection of MAGs/SAGs and reference genomes. Was a standard annotation pipeline used for all, or were pre-provided annotations used. It is helpful for the reader to know this without having to look through the methods.

This information was already detailed in the methods section entitled "Recalling Open Reading Frames".

11. It might have been interesting to test some of the antibiotic resistance related gene families that were noted – perhaps this should be suggested in the discussion (I think it is out of the scope of the current manuscript).

Yes, we agree that there are some promising predictions in our dataset, but, indeed, experimental assays for such cases are complex and out of the scope of this work.

12. Line 218 – don't think the word 'basis' is necessary.

Fixed

13. Line 327 – word choice: 'candidates'

Fixed

Referee #2 (Remarks to the Author):

The authors have fully addressed all my previous queries. I particularly enjoy the new search tool on their website and the clarity of words in sections I wasn't able to comprehend before. Congratulations to this great piece of work.

We thank the reviewer for his/her suggestions to improve the paper